# High-Dynamic Radar Sequence Prediction for Weather Nowcasting Using Spatiotemporal Coherent Gaussian Representation

**Ziye Wang**[1,2]*, **Yiran Qin**[2], **Lin Zeng**[3], **Ruimao Zhang**[1]†
[1] Sun Yat-sen University   [2] The Chinese University of Hong Kong, Shenzhen
[3] Guangzhou Meteorological Observatory

## Abstract

Weather nowcasting is an essential task that involves predicting future radar echo sequences based on current observations, offering significant benefits for disaster management, transportation, and urban planning. Current prediction methods are limited by training and storage efficiency, mainly focusing on 2D spatial predictions at specific altitudes. Meanwhile, 3D volumetric predictions at each timestamp remain largely unexplored. To address such a challenge, we introduce a comprehensive framework for 3D radar sequence prediction in weather nowcasting, using the newly proposed SpatioTemporal Coherent Gaussian Splatting (STC-GS) for dynamic radar representation and GauMamba for efficient and accurate forecasting. Specifically, rather than relying on a 4D Gaussian for dynamic scene reconstruction, STC-GS optimizes 3D scenes at each frame by employing a group of Gaussians while effectively capturing their movements across consecutive frames. It ensures consistent tracking of each Gaussian over time, making it particularly effective for prediction tasks. With the temporally correlated Gaussian groups established, we utilize them to train GauMamba, which integrates a memory mechanism into the Mamba framework. This allows the model to learn the temporal evolution of Gaussian groups while efficiently handling a large volume of Gaussian tokens. As a result, it achieves both efficiency and accuracy in forecasting a wide range of dynamic meteorological radar signals. The experimental results demonstrate that our STC-GS can efficiently represent 3D radar sequences with over $16\times$ higher spatial resolution compared with the existing 3D representation methods, while GauMamba outperforms state-of-the-art methods in forecasting a broad spectrum of high-dynamic weather conditions. Code and dataset are available on our project page.

## 1 Introduction

Weather nowcasting is a critical component of meteorological forecasting that focuses on predicting short-term weather conditions based on real-time observations. It supports many meteorological applications, such as precipitation forecasting Shi et al. (2017); Yu et al. (2024), extreme weather warnings Zhang et al. (2023), and hurricane prediction Li et al. (2022), and plays a vital role in various applications, including disaster management, transportation safety, and urban planning. Accurate nowcasting relies on timely and precise predictions of rapidly changing weather patterns. One of the key pathways for weather nowcasting is radar sequence prediction, which focuses on forecasting future radar echo frames from current observations.

Recent advancements have primarily concentrated on predicting 2D radar sequences at specific altitudes Wang et al. (2017); Gao et al. (2022b); Wang et al. (2023); Gao et al. (2023); Yu et al. (2024). However, the atmospheric system develops in three spatial dimensions, where the intensity of echoes from low-altitude clouds is closely linked to surface convective activity, while higher-altitude clouds

---

*Work completed by Ziye Wang as a Research Assistant under the supervision of Prof. Ruimao Zhang. ziye.wang1998@outlook.com
†Corresponding author is Prof. Ruimao Zhang. ruimao.zhang@ieee.org

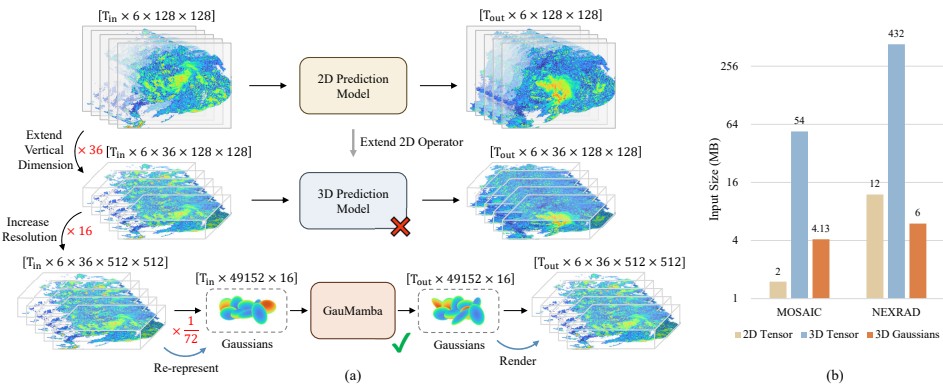

Figure 1: **Comparisons between 2D and 3D radar sequence prediction frameworks.** (a) Current 2D methods mainly focus on 2D spatial predictions at specific altitudes. Limited by training and storage efficiency, it is challenging to extend these methods to 3D architectures directly. (b) The size of the storage occupied by a single frame input is calculated according to the resolution of two datasets. The resolution of MOSAIC is $36 \times 384 \times 512$, 2D input is $384 \times 512$, 3D input is $36 \times 384 \times 512$, and the size of 3D Gaussians is $49,152 \times 11$. The resolution of NEXRAD is $36 \times 512 \times 512$ with 6 channels, the inputs are $6 \times 512 \times 512$, $6 \times 36 \times 512 \times 512$ and $49,152 \times 16$.

indicate potential severe weather. Relying solely on 2D radar echoes at specific altitudes overlooks the dependencies between different heights and fails to capture the complete structure of the system. As shown in Fig. 1, a straightforward way is to extend the 2D predictive model to 3D. However, this transition requires the model to receive higher-dimensional features, leading to increased memory usage and computational complexity, which further restricts high-resolution scaling.

To address such an issue, we present a comprehensive framework for effective 3D radar sequence prediction. This framework utilizes the newly proposed SpatioTemporal Coherent Gaussian Splatting (STC-GS) for dynamic radar representation and incorporates GauMamba for accurate and efficient forecasting. Specifically, our STC-GS differs from traditional methods that rely on 4D Gaussians for daily dynamic scene reconstruction, where only the main subjects are dynamic and most objects are treated as locally static. In contrast, radar observations of clouds are continuously in motion and changing, without adhering to any rigid body or morphological constraints. STC-GS begins by reconstructing the initial frame utilizing a group of 3D Gaussians (Kerbl et al., 2023), which are then employed as anchors. By monitoring and aggregating changes to these anchors as subsequent frames are pre-reconstructed, STC-GS effectively captures the underlying motion trends of the radar sequence. The initial Gaussians from the first frame, combined with the series of motion trends extracted from subsequent frames, establish a clear temporal continuity. Such a scheme ensures the consistent tracking of each 3D Gaussian over time, thereby enhancing its efficacy in predictive tasks. In practice, we develop a bidirectional reconstruction pipeline for STC-GS that accurately models the growth and dissipation of radar echoes. This pipeline incorporates dual-scale constraints to preserve the temporal coherence of Gaussian movements across both global trends and local details in the data.

Once we establish temporally correlated Gaussian groups across all of the frames, we adopt them to train the proposed GauMamba. It integrates a memory mechanism into the Mamba framework, allowing it to model the temporal evolution of Gaussian groups while efficiently managing a large volume of Gaussian tokens. This capability is crucial for effectively forecasting a wide range of dynamic meteorological radar signals. Overall, our framework not only enhances the accuracy of 3D radar sequence predictions but also improves computational efficiency, making it a robust solution for weather nowcasting applications. By leveraging the strengths of STC-GS and GauMamba, we aim to advance the state-of-the-art in meteorological forecasting, providing timely and accurate information for various practical applications.

The main contributions of this work can be summarized as follows. (1) To our best knowledge, this is the first work for 3D-based weather nowcasting by predicting high-dynamic radar sequences. Our framework introduces a novel 3D Gaussian representation termed STC-GS that adeptly captures radar data dynamics, paired with a memory-augmented network, GauMamba, which learns temporal evolution from these representations to forecast radar changes. (2) We propose a bidirectional

Gaussian reconstruction pipeline, which is designed to precisely track the motion trajectory of 3D Gaussians along the sequential frames. It incorporates dual-scale constraints to ensure coherence at both global trends and local details, effectively preserving the temporal consistency of Gaussian movements. (3) In addition, we collect and organize a novel high-dynamic 3D radar sequence dataset named MOSAIC and reorganize a dataset named NEXRAD. Our experiments demonstrate that GauMamba effectively predicts future 3D radar data with $16\times$ resolution, achieving a $19.7\%$ and $50\%$ reduction in Mean Absolute Error (MAE), and a notable improvement in predicting regions with significant radar signals.

## 2 RELATED WORK

### 2.1 3D GAUSSIAN REPRESENTATION

Recently, Gaussian Splatting-based representations offer real-time rendering with high training efficiency and garner considerable research interestsTang et al. (2023); Diolatzis et al. (2024); Mallick et al. (2024); Zhou et al. (2024). Motivated by the success of 3D Gaussian Splatting, numerous studies have extended it to real-time dynamic scene reconstruction and rendering. Incremental translation methods Luiten et al. (2023); Sun et al. (2024) tackle this challenge by initializing each frame based on the preceding one, leveraging motion constraints to enforce temporal coherence. Other approaches extend Gaussian representations to 4D space-time Yang et al. (2024a) or model global scene deformations with neural networks or polynomial Wu et al. (2024); Yang et al. (2024b); Li et al. (2024), enabling efficient reconstructions. Despite these advancements, such methods face limitations in accurately capturing complex or discontinuous scene dynamics. Moreover, they often rely on fixed color and opacity settings to ensure consistency.

Recent studies Shen et al. (2024); Yi et al. (2024); Ziwen et al. (2024); Zhang et al. (2025) have explored integrating Mamba Gu & Dao (2023); Dao & Gu (2024) or Transformer architectures with 3D Gaussian representations, focusing on reconstructing 3D Gaussians from single or multi-view images. However, these approaches are limited in their ability to model spatiotemporal dynamics, as the lack of memory mechanisms. Our method leverages sequences of 3D Gaussians to represent the temporal evolution of 3D radar echo data and employs a Memory-Augmented GauMamba model to effectively integrate information from preceding frames.

### 2.2 SPATIO-TEMPORAL PREDICTION

Spatio-temporal prediction is crucial in meteorological forecasting, requiring models to capture both spatial patterns and temporal dynamics. U-Net architectures using 2D or 3D CNNs have been applied to tasks like precipitation nowcasting, Arctic Sea ice prediction, and ENSO forecasting Veillette et al. (2020); Andersson et al. (2021); Ham et al. (2019), though they struggle with temporal dependencies. To improve this, methods such as ConvLSTM Shi et al. (2015), ConvGRU Shi et al. (2017), and PredRNN Wang et al. (2017; 2023) integrate memory mechanisms to better handle spatio-temporal correlations. E3D-LSTM Wang et al. (2019) combines 3D CNNs with LSTM for long-term forecasting, while PhyDNet Guen & Thome (2020) embeds physical constraints into models. SimVP Gao et al. (2022a) simplifies prediction using convolutional encoders and decoders, while transformer-based models Pathak et al. (2022); Bai et al. (2022); Gao et al. (2022b) capture long-range dependencies. However, deterministic models often struggle with prediction blur and fail to capture the stochastic nature of weather systems. To address this, diffusion-based models Gao et al. (2023); Yu et al. (2024) have been introduced to estimate spatio-temporal uncertainty.

## 3 METHODOLOGY

### 3.1 PRELIMINARY

**3D Gaussian Splatting (3DGS).** Gaussian Splatting Kerbl et al. (2023) uses a collection of 3D Gaussians to represent 3D objects or scenes. Each Gaussian is characterized by a position $\boldsymbol{p} \in \mathbb{R}^3$, a scaling factor $\boldsymbol{s} \in \mathbb{R}^3$, and a rotation quaternion $\boldsymbol{q} \in \mathbb{R}^4$. For rendering RGB images, an opacity value $\alpha \in \mathbb{R}$ and a color feature $\boldsymbol{c} \in \mathbb{R}^C$ are also included, with optional spherical harmonics for

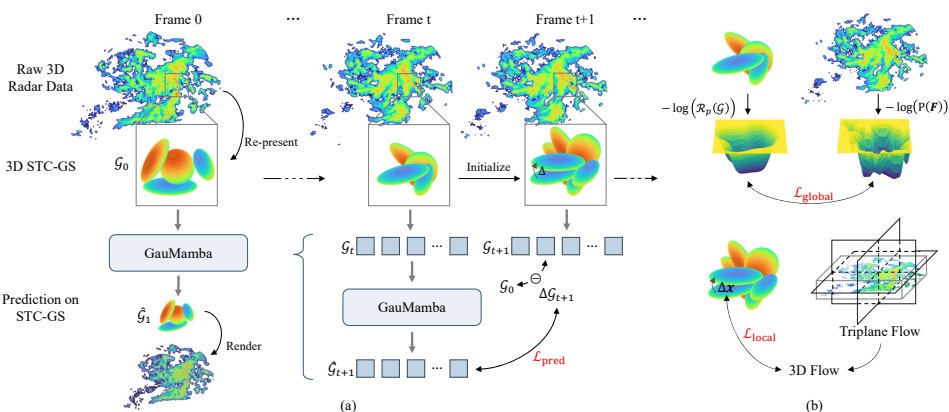

Figure 2: **Overview of our 3D prediction framework based on STC-GS and GauMamba**. (a) STC-GS can effectively compress the size of 3D data while fully representing it. GauMamba is a memory-augmented predictive model that leverages STC-GS for effective and accurate predictions. The STC-GS at Frame $t$ is input into the GauMamba to predict a set of DiffGaussians, $\Delta\mathcal{G}_{t+1}$, representing the differences between $\mathcal{G}_{t+1}$ and $\mathcal{G}_0$. This process is applied iteratively from Frame 0 to Frame $T_{in} + T_{out} - 1$. (b) In the process of radar reconstruction, dual-scale constraints are implemented to capture both the global trends and the local details present in the Gaussian motions.

modeling view-dependent effects. In this way, a group of Gaussians can be represented as $\mathbf{\Theta}$, where $\mathbf{\Theta}_i = \{\boldsymbol{p}_i, \boldsymbol{s}_i, \boldsymbol{q}_i, \alpha_i, \boldsymbol{c}_i\}$ denotes the parameters of the $i$-th Gaussian.

The rasterization stage involves calculating the Gaussian's contribution to individual pixels. Initially, each Gaussian is projected into the camera's coordinate. Following this, the renderer divides the screen into tiles and culls the Gaussians that fall outside the view frustum. Lastly, the renderer conducts alpha blending in a depth-sorted order within the view space for each pixel.

## 3.2 THE OVERALL FRAMEWORK

As illustrated in Fig. 2, we introduce a comprehensive framework for the prediction of 3D radar echo sequences in weather nowcasting. The overall framework includes two main components. The first part is the newly proposed SpatioTemporal Coherent Gaussian Splatting (STC-GS) for dynamic radar data representation. The second is the GauMamba, a memory-augmented Mamba network, coupled with STC-GS representations, for efficient and precise forecasting. In practice, unlike 3DGS, which is designed for rendering real-world RGB scenes, our STC-GS is specifically tailored for radar observations, whose definition is provided in Sec. 3.3.1.

Given the high-resolution 3D radar sequence, we first require to re-represent each frame $\boldsymbol{F}_i \in \mathbb{R}^{H \times W \times C}$ into a group of 3D Gaussians, where $H$ and $W$ donate the vertical and horizontal spatial resolution, and $C$ denotes the number of measurement channels. Unlike the traditional 3DGS technology used for scene reconstruction, our goal is to obtain a new representation of 3D radar data based on reconstruction to further accomplish downstream prediction tasks. Therefore, the proposed STC-GS should maintain the spatiotemporal coherence based on reconstruction. In this way, we introduce a bi-directional reconstruction scheme to achieve it, where a backward pre-reconstruction is first conducted to preliminarily retain the growth and dissipation of the Gaussians along the temporal dimension, and the forward reconstruction is carried out by using the pre-reconstruction results as the reference to enhance the accuracy. Please refer to Sec. 3.3.2 for more details.

Next, the prediction based on 3D radar can be transformed to predict the variation of STC-Gaussians along frames. We coupled the memory mechanism with Mamba to achieve linear time training and inference based on Gaussian Groups and the capacity to model the temporal evolution of Gaussian groups. Our proposed GauMamba takes parameters of $t$-th frame Gaussians $\mathcal{G}_t$ as inputs and predicts the difference between the $\mathcal{G}_{t+1}$ and $\mathcal{G}_0$, see 3.4 for more details.

## 3.3 SPATIOTEMPORAL COHERENT GAUSSIAN REPRESENTATION

The goal of reconstructing 3D radar data into a set of 3D Gaussians is to obtain a feature-dense representation that can nearly losslessly represent the original 3D radar data. To achieve this, we propose

a differentiable reconstruction pipeline motivated by 3D Gaussians Splatting (3DGS) method Kerbl et al. (2023). Instead of relying on Structure from Motion (SfM) in 3DGS, our STC-GS randomly samples points from the original 3D radar data to initialize the positions and radar features of the 3D Gaussians. We then design a differentiable radar profile renderer capable of rendering radar cross-sections from any angle, including horizontal, vertical, and oblique views. Readers can refer to Appx. A for more details.

### 3.3.1 DEFINITION OF 3D RADAR GAUSSIAN IN STC-GS

Given the specific location $\boldsymbol{x} \in \mathbb{R}^3$ of the radar frame, we use the slightly perturbated coordinates indices of the corresponding pixels as the initial positions of the 3D Gaussians, to avoid falling into saddle points in the optimization space. Unlike using RGB values to reconstruct real-world scenes, reconstructing radar observation data requires $N$ dimensional features $\boldsymbol{f} \in \mathbb{R}^N$, which indicates radar echo intensity, horizontal reflectivity factor, spectrum width, etc. The number of radar features $N$ depends on the specific radar dataset. Additionally, defining a 3D Gaussian requires an optimizable full 3D covariance matrix $\boldsymbol{\Sigma}$. It can be decomposed into a rotation matrix $\boldsymbol{R}$ and a scaling matrix $\boldsymbol{S}$ for by:

$$\boldsymbol{\Sigma} = \boldsymbol{R}\boldsymbol{S}\boldsymbol{S}^T\boldsymbol{R}^T$$

In practice, the rotation matrix $\boldsymbol{R}$ and the scaling matrix $\boldsymbol{S}$ are represented as a rotation quaternion $\boldsymbol{q} \in \mathbb{R}^4$ and a scaling factor $\boldsymbol{s} \in \mathbb{R}^3$, respectively. In summary, for the $i$-th Gaussian, the optimizable attributes are given by $\hat{\boldsymbol{\Theta}}_i = \{\boldsymbol{x}_i, \boldsymbol{f}_i, \boldsymbol{s}_i, \boldsymbol{q}_i\}$. We then optimize each Gaussian by minimizing the error between the rendered results and the radar profiles from the origin 3D radar data.

### 3.3.2 BIDIRECTION RECONSTRUCTION SCHEME.

Adding or removing Gaussians during reconstruction disrupts spatiotemporal consistency, making it impossible to track Gaussians throughout the sequence. However, simply disabling adaptive density control significantly degrades reconstruction quality, impairing the capture of cloud dynamics in radar sequences. To address these issues, we propose a bidirectional reconstruction strategy that preserves spatiotemporal consistency without altering the Gaussian set. This approach effectively captures frame-by-frame motion trends from raw data while accurately modeling the growth and dissipation of clouds in highly dynamic radar sequences.

The bidirectional reconstruction strategy consists of two stages: backward reconstruction and forward reconstruction. In the **backward reconstruction stage**, we pre-reconstruct from timestamp $T$ to timestamp $0$. Only the positions of the Gaussians are optimized using the **local detail and global trend constraints**. Components that persist or gradually dissipate over time are preserved, and each Gaussian's position is updated according to cloud motion at each timestamp. By the time backward reconstruction reaches timestamp $0$, it effectively incorporates information from future frames, bringing this information back to the initial frame. In the **forward reconstruction stage**, we iteratively reconstruct each frame from timestamp $0$ to timestamp $T$. We uniformly sample from the Gaussians obtained during the backward reconstruction and the original 3D data at timestamp $0$ to initialize a new set of Gaussians. In the coarse reconstruction stage, we optimize the positions using the local detail and global constraints, then optimize all Gaussian parameters by additional reconstruction loss.

Achieving a spatiotemporally coherent representation requires each Gaussian to precisely track the motion of the region it reconstructs. Besides, radar sequences often exhibit complex and diverse dynamics, including motion, deformation, growth, and dissipation. Reconstructing such high-dynamic sequences necessitates simultaneous adjustments to all attributes of the Gaussians. Identifying which parameters to modify for optimal error minimization becomes a significant challenge. Thus, the prediction based on 3D Gaussians introduces new challenges for the optimization process during the reconstruction phase. To address this, we propose a local detail constraint (3D flow constraint) and a global trend constraint (global energy constraint), which introduce 3D motion priors and optimal position distribution priors estimated from the original 3D radar data.

**Local Detail Constraint.** To introduce 3D motion priors from original 3D radar data and achieve a spatiotemporally coherent for each Gaussian, we introduce a 3D optical flow constraint, as shown in Fig. 2(b). First, we utilize a pre-trained 2D optical flow model, RAFT Teed & Deng (2020), to estimate the motion of radar observations across the xoy, xoz, and yoz planes. These 2D flows are

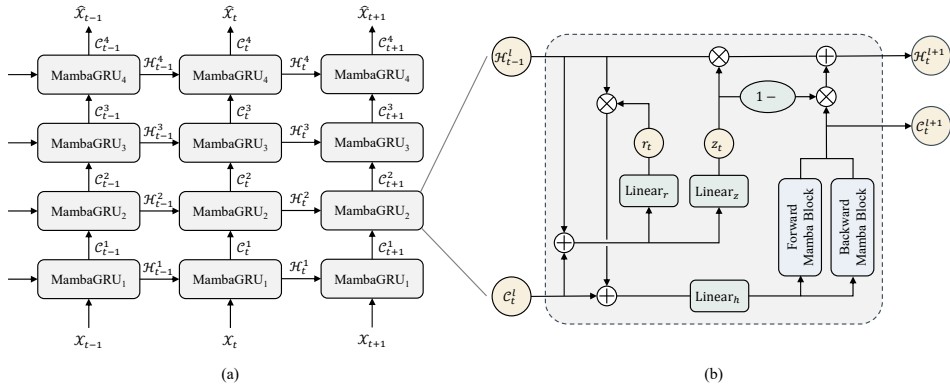

Figure 3: **The overall architecture of the GauMamba.** (a) Main architecture of GauMamba which consists of multiple stacked MambaGRU Block. (b) Detailed architecture design of MambaGRU.

then fused into a pseudo 3D flow within an xyz grid to approximate motion in 3D space. We take the 3D flow grid closest to each Gaussian as the reference flow, and constrain the distance between the displacement of the Gaussians $\Delta \boldsymbol{x} \in \mathbb{R}^3$ and the reference 3D flow $\Delta_{flow}$ to ensure that the Gaussians move in tandem with the corresponding cloud formations:

$$\mathcal{L}_{local} = \|\Delta \boldsymbol{x} - \Delta_{flow}\|^2.$$

However, since the 3D optical flow is estimated and fused from 2D images, it may not always be accurate. Moreover, employing an optical flow model pre-trained on real-world image sequences to estimate the flow of radar sequences could introduce inductive bias, which may accumulate over iterations. As a result, while the 3D flow constraint helps ensure that the Gaussians generally align with the direction of actual cloud motion, it cannot guarantee that each Gaussian is optimally positioned in every iteration.

**Global Trend Constraint.** From the original sparse 3D radar observation data, we can infer an optimal distribution of a group of Gaussians that accurately reconstructs the 3D data. Intuitively, the probability of Gaussians existing should be high in non-null regions of the 3D radar data, gradually decreasing to zero from the boundary areas between non-null and null regions to fully null regions, as shown in Fig. 2(b). We approximate the unnormalized optimal distribution of Gaussians using a smoothed function with threshold suppression. Specifically, we apply a Gaussian kernel-based smooth convolution to the original radar data and clamp values that exceed a predefined threshold $\tau$, the estimated unnormalized optimal distribution $\mathrm{P}(\boldsymbol{F})$ can be expressed mathematically as follows:

$$\mathrm{P}(\boldsymbol{F}) = \min(\boldsymbol{F} * \mathcal{K}(\cdot), \tau).$$

Here, $\boldsymbol{F}$ is the original radar data, $*$ denotes the convolution operator and $\mathcal{K}(\cdot)$ is the Gaussian kernel. For the current group of 3D Gaussians, we can derive their corresponding position probability densities by reusing the differentiable renderer. Specifically, each 3D Gaussian is treated as a normal distribution centered at its current location, and the renderer is used to obtain the position probability densities of the current Gaussians. Then, utilizing the energy function in energy-based learning LeCun et al. (2006), we can define a simple yet effective method to measure the similarity between two distributions:

$$\mathcal{L}_{global} = \|\mathcal{R}_{\mathrm{P}}(\mathcal{G}) - \mathrm{P}(\boldsymbol{F})\|^2.$$

Here, $\mathcal{R}_{\mathrm{P}}$ denotes the reused renderer for distribution estimation and $\mathcal{R}_{\mathrm{P}}(\mathcal{G})$ is the estimated distribution based on current Gaussians $\mathcal{G}$. By minimizing this energy function, we can ensure that each Gaussian in the current set is positioned at a more optimal initial location.

## 3.4 MEMORY AUGMENTED MAMBA PREDICTIVE MODEL

We now introduce our proposed Mamba-based prediction model termed GauMamba, incorporated with the spatiotemporal coherent Gaussian representation for high-dynamic 3D radar sequence prediction. After reconstructing the raw radar data into a sequence of Gaussian groups, we aim to train the predictive model based on the 3D Gaussian representations.

Thanks to STC-GS, the prediction of 3D radar sequence can be transformed into the prediction of variation of Gaussian group along the temporal dimension. We leverage the standard Mamba architecture Gu & Dao (2023); Dao & Gu (2024) and incorporate the GRU memory mechanism to

predict parameters of future Gaussians. The former can effectively handle a large number of Gaussian tokens as input, while the latter can effectively learn the spatiotemporal relationships between adjacent Gaussian Groups. As shown in Figure 3(a), the proposed GauMamba consists of multiple stacked MambaGRU blocks. The $l$-th layer MambaGRU block at timestamp $t$ takes two inputs: the output embedding from the previous layer $\mathcal{C}_t^{l-1}$ and the hidden state used for memory from the previous timestamp $\mathcal{H}_{t-1}^l$. The block then outputs the updated Gaussian embeddings $\mathcal{C}_t^l$, along with the updated memory hidden state $\mathcal{H}_t^l$ to be passed to the next timestamp.

**Training objective.** In our 3D Gaussians prediction training, we parameterize our model $f_\theta$ to predict DiffGaussian $\Delta\mathcal{G}_t = \mathcal{G}_t - \mathcal{G}_0$ for $t = 1, 2, \cdots, T_{in} + T_{out}$, using:

$$\mathcal{L}_{pred} = \sum_{t=1}^{T_{in+1}} \|f_\theta(\mathcal{G}_{t-1}, \mathcal{H}_{t-1}) - \mathcal{G}_t\|^2 + \sum_{t=T_{in+2}}^{T_{out}} \|f_\theta(\hat{\mathcal{G}}_{t-1}, \mathcal{H}_{t-1}) - \mathcal{G}_t\|^2$$

Here, $T_{in}$ is the length of given observation sequence, $T_{out}$ is the length of predicted sequence, and $\hat{\mathcal{G}}_{t-1} = \mathcal{G}_0 \oplus \Delta\hat{\mathcal{G}}_{t-1}$. Notably, since we only have access to true states of the first $T_{in}$ Gaussian Groups, there is no true value for timestamps beyond $T_{in+1}$ to be used as input for prediction. Thus, for $t \leq T_{in} + 2$, $\hat{\mathcal{G}}_{t-1}$ is estimated based on the predicted result $\Delta\hat{\mathcal{G}}_{t-1}$ from the previous timestamp and the initial state $\mathcal{G}_0$. Specifically, we tokenize $\mathcal{G}_0$ to obtain its embedding, and then combine it with the embedding of $\Delta\hat{\mathcal{G}}_{t-1}$, which is produced by the final MambaGRU block from the previous time step. This summation results in the estimated embedding for timestamp $t - 1$.

# 4 EXPERIMENT

More details of datasets, implementation, metrics, and experimental setup are in Appendix C.

## 4.1 DATASET

The datasets used in this study include NEXRAD and MOSAIC. **NEXRAD**[1] comprises radar observations of severe storms in the U.S., with 3D reflectivity data sampled at 5-minute intervals. Seven radar features, such as reflectivity, azimuthal shear, differential reflectivity, and so on, are included. **MOSAIC** records radar observations of storms in Guangdong, China, with 6-minute intervals, focusing solely on intensity data of radar echoes. Both datasets are preprocessed to ensure consistent vertical spacing and are divided into training, validation, and test sets. The prediction task involves forecasting up to 20 future frames based on 5 observed frames. For further information, please refer to the supplementary material C.1.

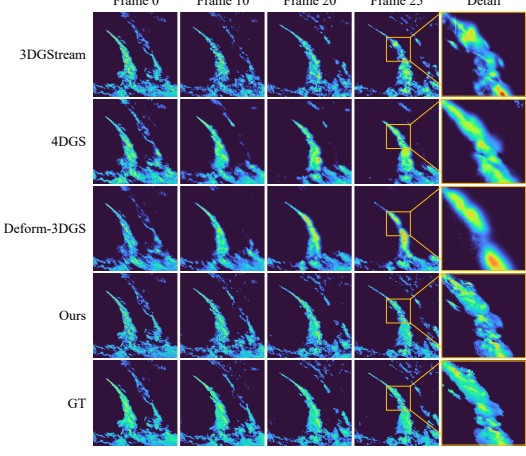

Figure 4: Qualitative results of reconstruction.

## 4.2 EXPERIMENT RESULTS

**Resconstruction.** First, we verify the effectiveness of our proposed re-represent method on the high-dynamic 3D radar dataset and compare it with other state-of-the-art methods designed for dynamic scene reconstruction. Table 1 presents the quantitative evaluation result on reconstructing hight-dynamic 3D radar sequences. The competing methods adhere to their original configurations, with certain parameters fixed during optimization. In contrast, our approach allows all parameters to be freely optimized, posing a significantly more challenging task. Despite this, our method achieves superior performance across all metrics. As illustrated in Figure 4, our proposed method preserves more details and maintains consistent accuracy throughout the entire sequence.

---

[1]https://huggingface.co/datasets/Ziyeeee/3D-NEXRAD

Table 1: Comparision of reconstruction in NEXRAD

| Model | $\text{MAE}^{\downarrow}_{\times 10}$ | $\text{PSNR(dB)}^{\uparrow}$ | $\text{SSIM}^{\uparrow}$ | $\text{LPIPS}^{\downarrow}$ | $\text{LPIPS}^{\downarrow}_{\text{Radar}}$ |
|---|---|---|---|---|---|
| 3DGStream Sun et al. (2024) | 0.019 | 38.133 | 0.954 | 0.091 | 0.902 |
| 4DGS Wu et al. (2024) | 0.028 | 35.731 | 0.933 | 0.135 | 0.623 |
| Deform-3DGS Yang et al. (2024b) | 0.029 | 35.027 | 0.931 | 0.141 | 0.578 |
| Ours | **0.014** | **40.262** | **0.970** | **0.057** | **0.123** |

**Prediction.** The large and cumbersome 3D tensors present a significant challenge in extending radar echo predictions from 2D to 3D, thereby limiting the scalability of models for high-resolution predictions. Fig. 5 illustrates the GPU memory requirements for feature extraction and prediction using methods based on 3D data, including ConvGRU, PhyDNet, SimVP, and DiffCast, alongside our proposed feature-dense 3D Gaussian prediction approach. Scaling these radar echo prediction methods, which use raw 3D data as input, to high-resolution training and deployment is both difficult and impractical, as the time and space complexity scales quadratically, *i.e.*, $O(\text{N}^2)$, with respect to the horizontal resolution of $\text{N} \times \text{N}$. In contrast, the memory usage of our

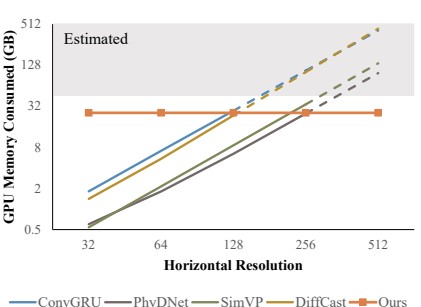

Figure 5: Memory usage of different methods with various input resolutions.

proposed Gaussian-based method is independent of the input or output resolution, scaling linearly only with the number of Gaussian primitives. As shown in Fig. 5, the memory usage of our Gau-Mamba method is evaluated with a fixed number of Gaussian primitives tailored for high-resolution images, balancing computational efficiency and reconstruction precision. When predicted at lower resolutions, e.g., $\text{N} \leq 256$, fewer Gaussian primitives are required, leading to reduced memory consumption.

With the available computational resources of 4 A100 80G GPUs, methods that use raw 3D data as input cannot be effectively trained at a horizontal resolution of $512 \times 512$. To facilitate a fair comparison with our proposed GauMamba, we trained ConvGRU, PhyDNet, SimVP, and DiffCast at a horizontal resolution of $128 \times 128$, and trained GauMamba at $512 \times 512$, while maintaining a consistent vertical resolution. Then, we upsampled the predictions from ConvGRU, PhyDNet, SimVP, and DiffCast to $512 \times 512$ for a fair evaluation.

The quantitative performance metrics of two 3D radar datasets are presented in Table 2 and 3 respectively. In general, our proposed GauMamba with STC-GS as input outperforms all existing models based on raw 3D radar data, even if predicting in an actual higher resolution. The Gau-Mamba achieves 12.1%, 69.0% and 4.8%, 101.1% improvements in CSI-20 and CSI-30 with $4 \times 4$ pooling in MOSAIC and NEXRAD dataset, respectively. We also present the qualitative results in Fig. 6. The prediction results of our proposed method contain more details and the high values of the significant regions are preserved. This is because the re-represented Gaussian with significant features is more prioritized to be tracked and predicted in order to minimize our optimization objective. In addition, the results of Mamba and our GauMamba demonstrate that the approaches and experiments developed within our proposed framework, which redefine the 3D prediction task by first re-representing the sequences with 3D Gaussians and then predicting their future evolution, significantly outperform traditional methods, highlighting the effectiveness and robustness of our reformulation.

## 4.3 ABLATION STUDY

### 4.3.1 RECONSTRUCTION

**Reconstruction w/o local details.** The flow constraint introduces a pseudo-3D motion flow that guides the reconstruction process. Its absence leads to dynamic Gaussians that struggle to accurately track the movement of the reconstructed elements. This deficiency results in a 32.6% increase in

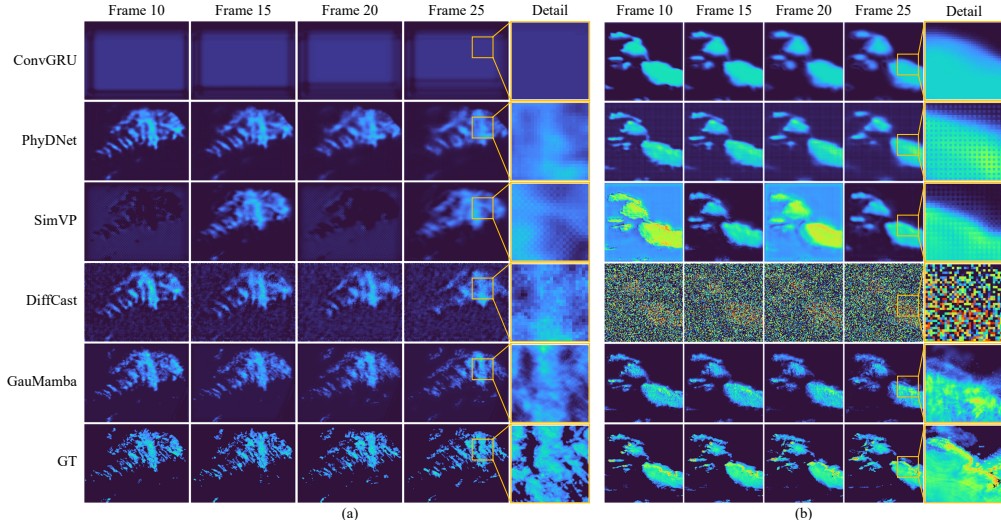

Figure 6: **Qualitative results of 3D radar prediction.** (a) The prediction results in MOSAIC dataset. (b) The prediction results in NEXRAD dataset. Detail comparisons are plotted in the last column, respectively. Our proposed framework can more effectively predict local changes in detail, which may determine specific weather change trends.

Table 2: Experiment results in MOSAIC

| Model | MAE$^\downarrow$ | SSIM$^\uparrow$ | LPIPS$^\downarrow$ | LPIPS$^\downarrow_{Radar}$ | CSI-20$^\uparrow_{Pool4}$ | CSI-30$^\uparrow_{Pool4}$ | CSI-40$^\uparrow_{Pool4}$ |
|---|---|---|---|---|---|---|---|
| ConvGRU | 1.728 | 0.621 | 0.303 | 4.837 | - | - | - |
| PhyDNet | 0.910 | 0.810 | 0.244 | 1.451 | 0.294 | 0.108 | 0.002 |
| SimVP | 0.890 | 0.835 | 0.270 | 3.516 | 0.264 | 0.075 | - |
| DiffCast | 1.878 | 0.355 | 0.433 | 2.216 | 0.305 | 0.126 | 0.006 |
| Mamba | 0.750 | 0.894 | 0.164 | 0.777 | 0.293 | 0.166 | 0.055 |
| GauMamba | **0.714** | **0.897** | **0.157** | **0.741** | **0.342** | **0.213** | **0.062** |

the MAE score. **Reconstruction w/o global trends.** The energy-based constraint plays a crucial role during the initial stages by ensuring that the set of Gaussians achieves a more appropriate coarse-level spatial distribution. This constraint also helps mitigate error accumulation that can arise from inaccurate flow estimations. After removing the energy-based iteration phase, we keep the total number of iterations constant. **Reconstruction w/o bidirectional reconstruction.** Before the forward reconstruction phase, a backward reconstruction is conducted to optimize the spatial distribution of the Gaussians. Omitting this reverse pre-reconstruction stage results in the loss of the ability to propagate information about areas that emerge in the future back to the first frame. Consequently, the initialization of Gaussian positions becomes inadequate, leading to a significant decrease in accuracy.

### 4.3.2 PREDICTION

**GauMamba w/o Mermory.** Removing the memory mechanism results in a vanilla Mamba, which is not suitable for the current prediction task. To evaluate the effectiveness of the memory mechanism, we incorporate DLinear Zeng et al. (2023), a simple yet effective prediction method that does not rely on memory. Specifically, we split a vanilla Mamba into two symmetric parts: an encoder and a decoder. The encoder generates embeddings from historical observations, which are subsequently fed into DLinear to predict future embeddings. The decoder then produces the predicted parameters of Gaussian. However, Mamba with DLinear does not surpass our method with a 20.2% increment of MAE score. Our memory mechanism can be updated temporally while considering global embeddings. In contrast, DLinear infers future embeddings based solely on local

Table 3: Experiment results in NEXRAD

| Model | MAE$^\downarrow$ | SSIM$^\uparrow$ | LPIPS$^\downarrow$ | LPIPS$_{Radar}^\downarrow$ | CSI-20$_{Pool4}^\uparrow$ | CSI-30$_{Pool4}^\uparrow$ | CSI-40$_{Pool4}^\uparrow$ |
|---|---|---|---|---|---|---|---|
| ConvGRU | 0.006 | 0.819 | 0.205 | 1.621 | 0.306 | - | - |
| PhyDNet | 0.017 | 0.373 | 0.320 | 2.058 | 0.311 | 0.089 | 0.002 |
| SimVP | 0.066 | 0.379 | 0.481 | 2.925 | 0.085 | 0.088 | 0.018 |
| DiffCast | 0.157 | 0.004 | 0.932 | 4.057 | 0.049 | 0.021 | 0.021 |
| Mamba | 0.004 | 0.899 | 0.129 | 0.699 | 0.309 | 0.165 | 0.074 |
| GauMamba | **0.003** | **0.900** | **0.126** | **0.665** | **0.326** | **0.179** | **0.078** |

Table 4: Ablation Study in Reconstruction

| Method | MAE$_{\times 100}^\downarrow$ | PSNR(dB)$^\uparrow$ | SSIM$^\uparrow$ | LPIPS$^\downarrow$ |
|---|---|---|---|---|
| Reconstruct w/o flow (local details) | 1.947 | 37.521 | 0.953 | 0.089 |
| Reconstruct w/o energy (global trends) | 1.497 | 40.049 | 0.969 | 0.059 |
| Reconstruct w/o bi-direction | 3.245 | 32,746 | 0.921 | 0.152 |
| Full Model | **1.468** | **40.262** | **0.970** | **0.057** |

Table 5: Ablation Study in Prediction

| GauMamba | ME$^{\rightarrow 0}$ | MAE$^\downarrow$ | SSIM$^\uparrow$ | LPIPS$^\downarrow$ | CSI-20$_{Pool4}^\uparrow$ | CSI-30$_{Pool4}^\uparrow$ | CSI-40$_{Pool4}^\uparrow$ |
|---|---|---|---|---|---|---|---|
| w/o Memory | -0.445 | 0.858 | 0.883 | 0.170 | 0.224 | 0.122 | 0.035 |
| w/o GRU | -0.374 | 0.743 | 0.895 | 0.161 | 0.289 | 0.165 | 0.052 |
| w/o Sort | -0.532 | 0.880 | 0.883 | 0.183 | 0.179 | 0.085 | 0.025 |
| Full Model | **-0.103** | **0.714** | **0.897** | **0.157** | **0.342** | **0.213** | **0.062** |

embeddings. **GauMamba w/o GRU.** We removed the reset gate and update gate from the GRU, while retaining the memory mechanism. The hidden states, which store historical information, are directly concatenated with the current Gaussian embeddings. In this configuration, the MAE score increases by $4.1\%$, which is lower than that of GauMamba w/o Mermory, highlighting the effective-ness of the memory mechanism. However, removing GRU reduces the flexibility in updating the memory. **GauMamba w/o Sort.** Morton sorting aggregates spatially adjacent points in the 1D se-quence. Without Morton sorting, the Gaussians are input into GauMamba in their original unsorted order, as determined during the reconstruction stage. Although each Gaussian maintains its position consistently across frames, a significant performance degradation is observed. This suggests that spatially unstructured sequences negatively impact the feature extraction capabilities.

## 5 CONCLUSION

In this work, we introduced a novel framework for 3D radar sequence prediction in weather fore-casting, addressing the limitations of current 2D spatial prediction methods. Our proposed Spa-tioTemporal Coherent Gaussian Splatting (STC-GS) and GauMamba effectively capture dynamic radar signals while maintaining high efficiency and accuracy. Experimental results demonstrate that STC-GS achieves superior reconstruction accuracy compared to existing 3D representations, while GauMamba outperforms state-of-the-art models in predicting dynamic weather conditions. In general, we provide a general solution framework for 3D prediction, paving the way for future advancements in this domain. However, the current scope of our approach is specialized, focusing primarily on radar-based weather nowcasting, with limited exploration of broader applications. We are actively working to enhance its generalization and simplify the pipeline for broader applicability to dynamic 3D scenarios.

## ACKNOWLEDGEMENT

This work is partially supported by the Natural Science Foundation of Guangdong Province (2022A1515011346), the Young Scientists Fund of the National Natural Science Foundation of China (62106154), Shenzhen Science and Technology Program (JCYJ20220818103001002), the Guangdong Key Laboratory of Big Data Analysis and Processing, Sun Yat-sen University, China, and the High-performance Computing Public Platform (Shenzhen Campus) of Sun Yat-sen University. We thank to Huaiyu Li from Guangzhou Meteorological Observatory and Haowen Li from Quzhou Meteorological Bureau for their valuable support in the MOSAIC dataset.

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

## A   3D RADAR CROSS-SECTION RENDERING

3D Radar Cross-Section Rendering aims to project 3D Gaussians from the world coordinate system onto the virtual camera imaging plane to render specific cross-section results. First, we apply a transformation matrix to convert the 3D Gaussians from the world coordinates to the camera coordinates. Then, we discard any Gaussians that exceed a predefined threshold distance from the virtual imaging plane. Finally, the remaining Gaussians are rendered in order of their distances to the imaging plane, from nearest to farthest. The 3D Gaussian's contribution to the pixel can be calculated by:

$$F = \sum_{i \in N_p} \boldsymbol{f}_i \cdot e^{-\frac{1}{2}(\boldsymbol{x}_p - \boldsymbol{x}_{g_i})^T \boldsymbol{\Sigma}_i^{-1}(\boldsymbol{x}_p - \boldsymbol{x}_{g_i})}$$

where $N_p$ is the number of Gaussians overlapping with the given pixel, $\boldsymbol{f}_i$ is the radar features of $i$-th Gaussian, $\boldsymbol{x}_p$ and $\boldsymbol{x}_{g_i}$ are the centered positions of the pixel and $i$-th gaussian respectively, $\boldsymbol{\Sigma}_i^{-1}$ is the 3D covariance matrix of $i$-th Gaussian. The rasterization strategy, similar to the origin 3DGS, is fully differentiable, enabling to take full use of the GPU acceleration framework to optimize parameters of Gaussians using stochastic gradient descent.

## B   MEMORY MECHANISM.

The core of the Gated Recurrent Unit (GRU) is based on two primary gates: the reset gate and the update gate, which control the flow of information. Unlike LSTM, GRU combines the forget and input gates into a single update gate, simplifying the architecture. When a new input $\mathcal{C}_t$ is received at time step $t$, the reset gate $r_t$ controls how much of the previous hidden state $\mathcal{H}_{t-1}$ should be ignored when computing the candidate hidden state $\hat{\mathcal{H}}_t$. A smaller reset gate value causes the model to "forget" parts of the previous hidden state, effectively resetting the memory. The update gate $z_t$ determines how much of the previous hidden state $\mathcal{H}_{t-1}$ should be retained, controlling the balance between preserving past information and incorporating new data. the update gate $z_t$ controls the final hidden state $\mathcal{H}_t$, which is a combination of the previous hidden state $\mathcal{H}_{t-1}$ and the candidate hidden state $\hat{\mathcal{H}}_t$. The advantages of GRU's gating mechanism are that it allows efficient control over information flow and helps to mitigate the vanishing gradient problem by maintaining long-term dependencies more effectively.

As illustrated in Fig. 3, incorporating Mamba and GRU preserves the linear-time complexity of Mamba and introduces a long-range memory mechanism that allows mamba-based models to predict the next Gaussians based on the previous Gaussians. The key equations are shown below:

$$r_t = \sigma(\boldsymbol{W}_{cr}\mathcal{C}_t + \boldsymbol{W}_{hr}\mathcal{H}_{t-1} + \boldsymbol{b}_r)$$
$$z_t = \sigma(\boldsymbol{W}_{cz}\mathcal{C}_t + \boldsymbol{W}_{hz}\mathcal{H}_{t-1} + \boldsymbol{b}_z)$$
$$\hat{\mathcal{H}}_t = \boldsymbol{W}_{ch}\mathcal{C}_t + \boldsymbol{W}_{hh}(r_t \circ \mathcal{H}_{t-1}) + \boldsymbol{b}_h$$
$$\hat{\mathcal{C}}_t = \text{BiMamba}(\hat{\mathcal{H}}_t)$$
$$\mathcal{H}_t = z_t \circ \mathcal{H}_{t-1} + (1 - z) \circ \hat{\mathcal{C}}_t$$

## C   EXPERIMENT SETTING

### C.1   DATASET

**NEXRAD:** The 3D gridded radar reflectivity data used in this study were collected by the U.S. NEXRAD WSR-88D radar network. These data are sourced from the ds841.6 dataset product available through the National Center for Atmospheric Research (NCAR) Research Data Archive Department of Atmospheric Sciences, Texas A&M University & School of Meteorology, University of Oklahoma (2021). Due to constraints on storage and GPU resources, radar observations of severe storm events in 2022 with a longitude and latitude grid size ranging from 512 to 1024 are selected from different geographical coverage.

The selected storm events are observed at 5-minute intervals with a horizontal resolution of approximately 0.021 degrees. A total of 6255 3D radar observations are considered in this study. To

preprocess the raw radar data, a quality control procedure is applied, following the methodology outlined in Homeyer & Bowman (2022). Subsequently, a central $512 \times 512$ longitude/latitude grid is cropped from each event observation. In the vertical dimension, there are 28 levels, spanning from 0.5 km to 7 km with 0.5 km intervals, and from 7 km to 22 km with 1 km intervals. To ensure consistent vertical spacing of features, we interpolate the original data to obtain 44 vertical layers with 0.5 km intervals spanning from 0.5 km to 22 km. The 3D radar data from these storm events include seven channels: (1) $Z_H$, the horizontal reflectivity factor, which indicates the intensity of radar returns from precipitation; (2) $SW$, the spectrum width, representing the variability of Doppler velocities within the radar pulse volume; (3) $AzShr$, the azimuthal shear, a measure of wind shear in the horizontal plane often used to detect rotation within storm systems; (4) $Div$, the divergence, which reflects the horizontal divergence or convergence of wind fields; (5) $Z_{DR}$, the differential reflectivity, used to differentiate precipitation types by comparing horizontal and vertical polarizations; (6) $K_{DP}$, the specific differential phase, providing information on phase shifts of the radar signal, useful for estimating rainfall rates; and (7) $r_{HV}$, the correlation coefficient between horizontal and vertical polarizations, which assesses the uniformity of precipitation particles and identifies non-meteorological targets.

Given the objective of predicting up to 20 future frames (100 minutes) based on 5 observed frames (25 minutes), we sample 25 continuous frames with a stride of 10 from each event. These sequences are then divided into training, validation, and test sets in a 9:0.5:0.5 ratio.

**MOSAIC** The MOSAIC dataset records radar echoes over several years within Guangdong, China, with a time interval of 6 minutes between consecutive frames. To ensure efficient training while considering GPU and storage limitations, we excluded radar data that did not capture meteorological events, focusing on severe storm events in 2022. This resulted in a total of 24,542 radar observations. The raw data have a horizontal resolution of $880 \times 1050$ pixels, with 21 vertical layers ranging from 0.5 km to 6 km with 0.5 km intervals, from 6 km to 10 km with 1 km intervals, and additional layers at 12 km, 14 km, 15.5 km, 17 km, and 19 km. To achieve consistent vertical spacing of features, we interpolate the original data to obtain 38 vertical layers with 0.5 km intervals spanning from 0.5 km to 19 km. The central $768 \times 1024$ region is cropped and downsampled to $384 \times 512$, resulting in radar observation data of dimensions $38 \times 384 \times 512$. Unlike the NEXRAD dataset, MOSAIC includes only the intensity data of radar echoes. For the task of predicting up to 20 future frames (120 minutes) based on 5 observed frames (30 minutes), we sample 25 continuous frames with a stride of 20 from each event. These sequences are then divided into training, validation, and test sets in a 9:0.5:0.5 ratio.

## C.2 IMPLEMENTATION DETAILS

We define the prediction task as predicting 20 frames future frames based on 5 initial frames, following Yu et al. (2024). For reconstruction, we random sample $10\%$ points from non-null regions to initialize the Gaussians. To maintain a constant total number of Gaussians, additional points are randomly sampled from the null regions, bringing the final count to 49,152 Gaussians. During the reverse pre-reconstruction stage, we perform 5,000 iterations to adjust the positions of the Gaussians, applying 3D flow and energy constraints. In the forward reconstruction stage, we first optimize the positions of the Gaussians in the same manner for the first 5,000 iterations. Afterward, all Gaussian parameters are optimized, incorporating the 3D flow, energy constraints, and reconstruction loss. The initial learning rate is set to 0.002, which is gradually reduced to 0.0002 by the end of training. For the prediction task, we train our framework for 50 epochs using the Adam optimizer with a learning rate of 0.0005. For baseline methods, their 2D operators are extended to 3D to handle 3D radar prediction tasks, with their configurations adjusted accordingly for different datasets. All experiments are conducted under the same settings using 4 A100 GPUs.

## C.3 METRICS

To evaluate the accuracy of predictions, we calculate the Mean Error (ME) and Mean Absolute Error (MAE) to assess the overall numerical discrepancy between the predicted results and the ground truth. An ME value closer to zero indicates better accuracy; an ME greater than zero signifies that the predictions generally exceed the ground truth, while an ME less than zero indicates the opposite. To evaluate the visual quality of the predictions, we employ the Structural Similarity In-

Table 6: Extended Experiment results in NEXRAD

| Model | MAE$^\downarrow$ | SSIM$^\uparrow$ | LPIPS$^\downarrow$ | LPIPS$_{Radar}^\downarrow$ | CSI-20$_{Pool4}^\uparrow$ | CSI-30$_{Pool4}^\uparrow$ | CSI-40$_{Pool4}^\uparrow$ |
|---|---|---|---|---|---|---|---|
| ConvGRU | 0.006 | 0.836 | 0.194 | 1.632 | 0.326 | - | - |
| PhyDNet | 0.017 | 0.366 | 0.323 | 2.114 | 0.348 | 0.097 | 0.002 |
| SimVP | 0.008 | 0.817 | 0.176 | 1.483 | 0.227 | 0.002 | 0.000 |
| DiffCast | 0.152 | 0.005 | 0.925 | 4.005 | 0.051 | 0.023 | 0.044 |
| Mamba | 0.004 | 0.902 | 0.125 | 0.625 | 0.304 | 0.158 | 0.075 |
| GauMamba | **0.003** | **0.907** | **0.122** | **0.600** | **0.361** | **0.205** | **0.089** |

dex (SSIM)Wang et al. (2004), Peak Signal-to-Noise Ratio (PSNR), and Learned Perceptual Image Patch Similarity (LPIPS)Zhang et al. (2018). Here, we use the pretrained AlexNet as the evaluator of LPIPS. Besides, we pretrained a BiGAN model on radar data in a self-supervised manner. The LPIPS score calculated with the encoder of BiGAN is marked as LPIPS$_{Radar}$. The results in Table 2 and 3 demonstrate that LPIPS$_{Radar}$ is well-aligned with the original LPIPS results. More importantly, it highlights perceptual differences that were undetected by the original LPIPS and robust to noise. In Table 2, the scores for ConvGRU and DiffCast show discrepancies between LPIPS and LPIPS$_{Radar}$. Referring to Figure 6 left, it can be clearly observed that ConvGRU fails to predict the next few frames accurately, instead providing a smoothed average result. In contrast, DiffCast produces results that are closer to the ground truth but with some noise. The higher LPIPS score for DiffCast indicates that LPIPS lacks robustness to noise in radar data, while LPIPS$_{Radar}$ more accurately reflects the perceptual differences between the two methods.

In addition, the Critical Success Index (CSI) is used to quantify the degree of pixel-wise agreement between the prediction and the ground truth. CSI is defined as $\frac{Hits}{Hits + Misses + False\ Alarms}$, where Hits (truth=1, pred=1), Misses (truth=1, pred=0), and False Alarms (truth=0, pred=1) are counted after binarizing the continuous values of predictions and ground truth into 0/1 values at thresholds [20, 30, 40] dBz. Following Ravuri et al. (2021); Gao et al. (2023); Yu et al. (2024), we report the CSI at pooling scale $4 \times 4$, which relaxes the pixel-wise matching to evaluate the accuracy on neighborhood aggregations.

## C.4 PREDICTION

The existing prediction models used for comparison are originally designed for 2D sequence prediction. Therefore, adjustments must be made to adapt them for 3D radar sequence prediction. Two feasible modifications can be considered. The first involves treating the newly introduced third dimension, which represents the vertical axis, as a channel dimension. However, this approach would limit the model's ability to predict changes along the vertical axis as effectively as it does in the horizontal plane. Instead, we opted to extend these models from 2D to 3D, enabling them to natively handle 3D prediction. Specifically, we expanded 2D CNNs to 3D CNNs and similarly extended modules such as ConvGRU and ConvLSTM included in PhyDNet and DiffCast from 2D to 3D. Additionally, the 2D PDE constraint in PhyDNet is also adapted for 3D prediction.

## D EXTENDED EXPERIMENTS

We conduct extended experiments using the NEXRAD dataset spanning three years (2020–2022) under the same training epochs, aiming to investigate the impact of increased dataset diversity and size on model performance. As shown in Table 6, most metrics of these models exhibit slight improvements when trained on the extended dataset. This performance gain can be attributed to the enhanced data diversity and the increased number of iteration steps enabled by the larger dataset. Notably, our model continues to outperform others, demonstrating its robustness and effectiveness even under extended experimental conditions.

# E    QUALITATIVE RESULTS

**Reconstruction.** The full reconstructed results are shown in Fig. 7, 8 and 9.

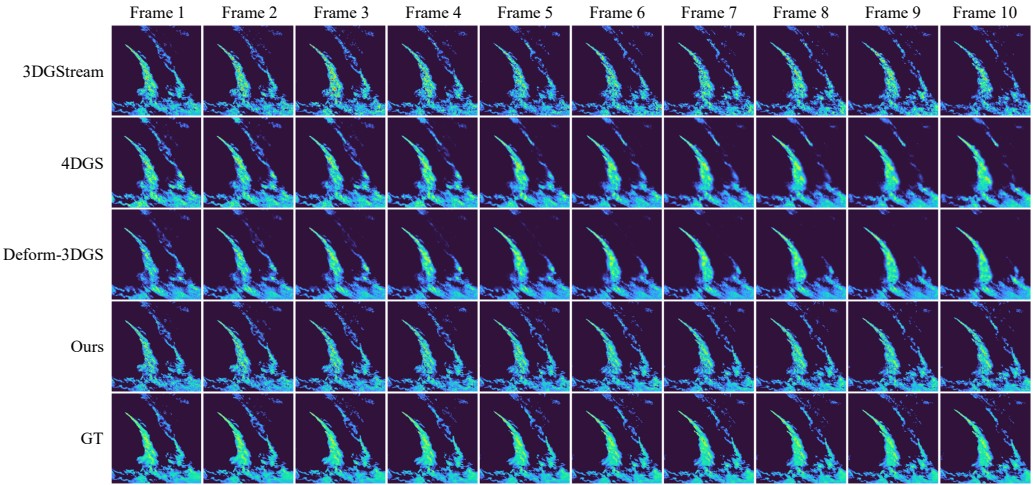

Figure 7: Qualitative results of 3D radar reconstruction from frame 1 to frame 10.

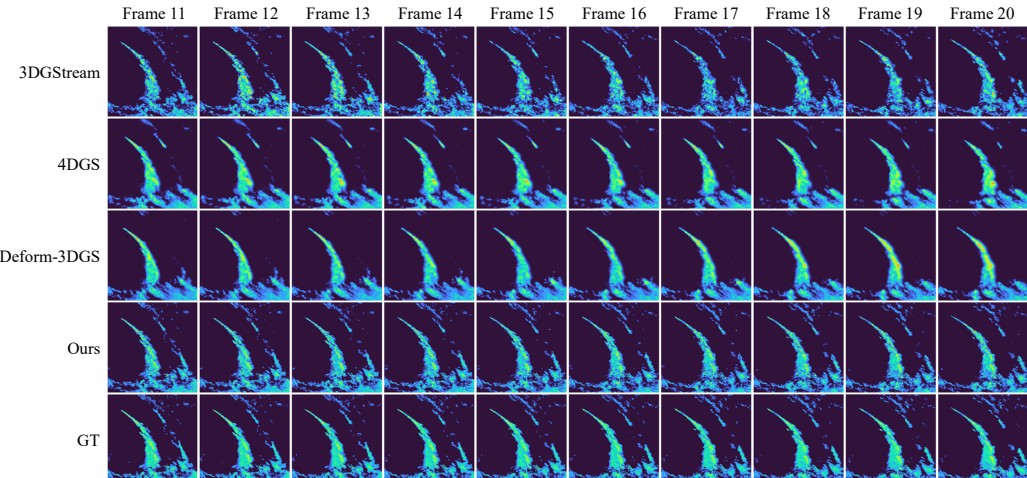

Figure 8: Qualitative results of 3D radar reconstruction from frame 11 to frame 20.

**Prediction.** The full predicted results are shown in Fig. 10, Fig. 11, Fig. 12, and Fig. 13

# F    DISCUSSION

**Training objective of GauMamba.** It is possible to directly predict $\mathcal{G}_t$ based on $\mathcal{G}_{t-1}$. However, we found that this approach often leads to poor convergence. The difficulty arises because directly predicting $\mathcal{G}_t$ requires the model to estimate both the overall trend of $\mathcal{G}$ and its exact position. In contrast, predicting $\Delta\mathcal{G}_t$ simplifies the problem by focusing only on the changes from the initial timestamp 0 to timestamp $t$, thus reducing the solution space and making it more compact. Another viable alternative is to predict the difference between consecutive timestamps, i.e., $\Delta\hat{\mathcal{G}}_t = \mathcal{G}_t - \mathcal{G}_{t-1}$. However, this method suffers from the drawback of error accumulation during inference, where the iterative predictions lead to increasingly compounded errors over time.

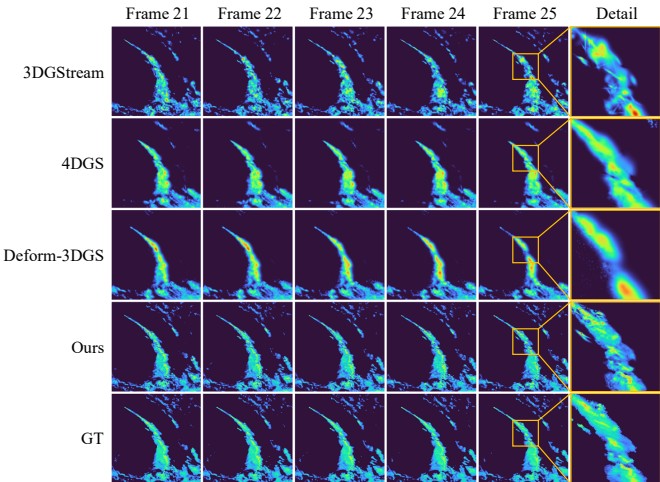

Figure 9: Qualitative results of 3D radar reconstruction from frame 21 to frame 25 and details of frame 25.

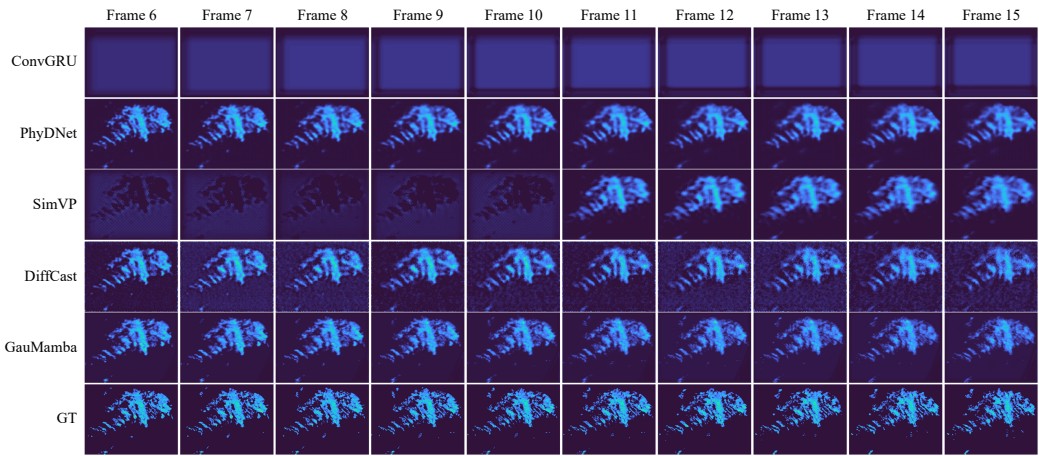

Figure 10: Qualitative results of 3D radar prediction from frame 6 to frame 15 in MOSAIC dataset.

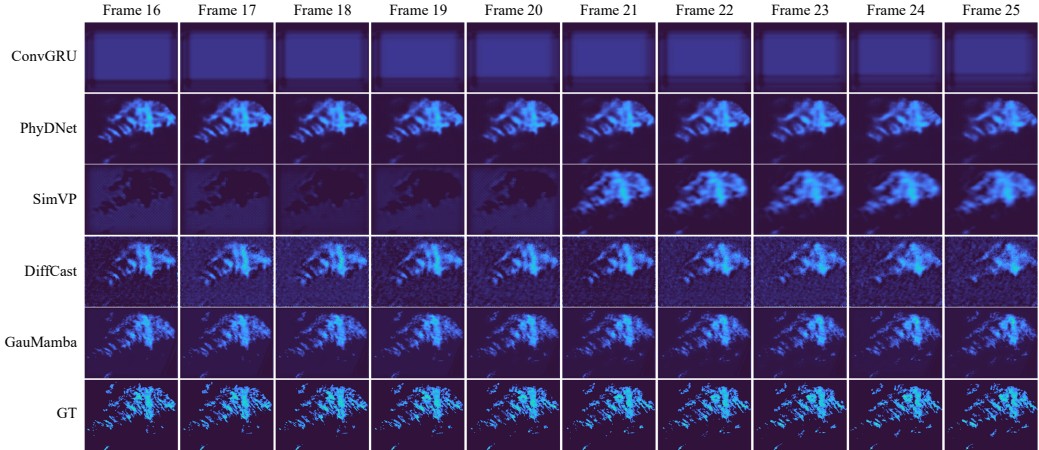

Figure 11: Qualitative results of 3D radar prediction from frame 16 to frame 25 in MOSAIC dataset.

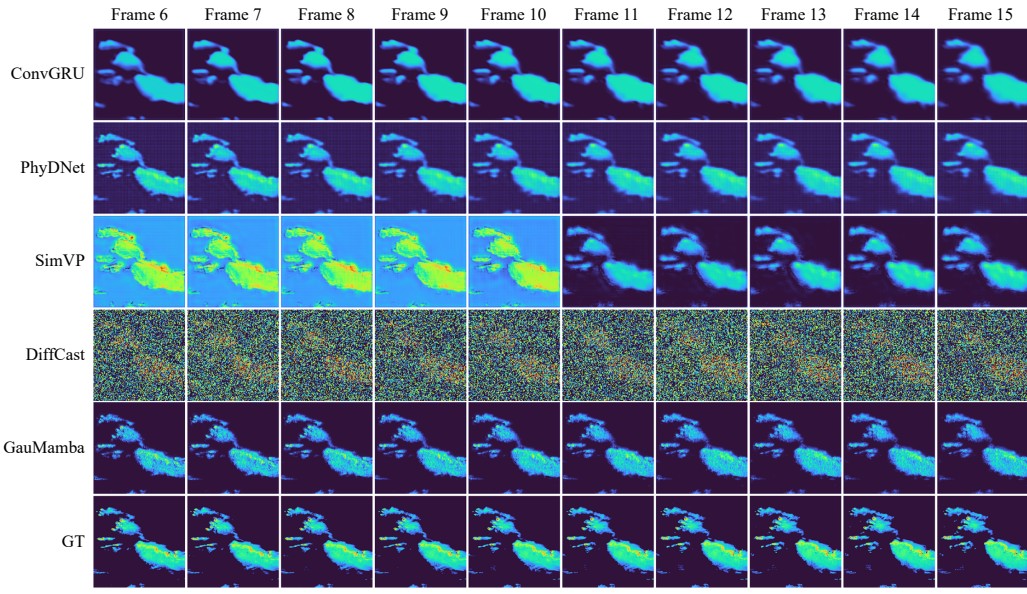

Figure 12: Qualitative results of 3D radar prediction from frame 6 to frame 15 in NEXRAD dataset.

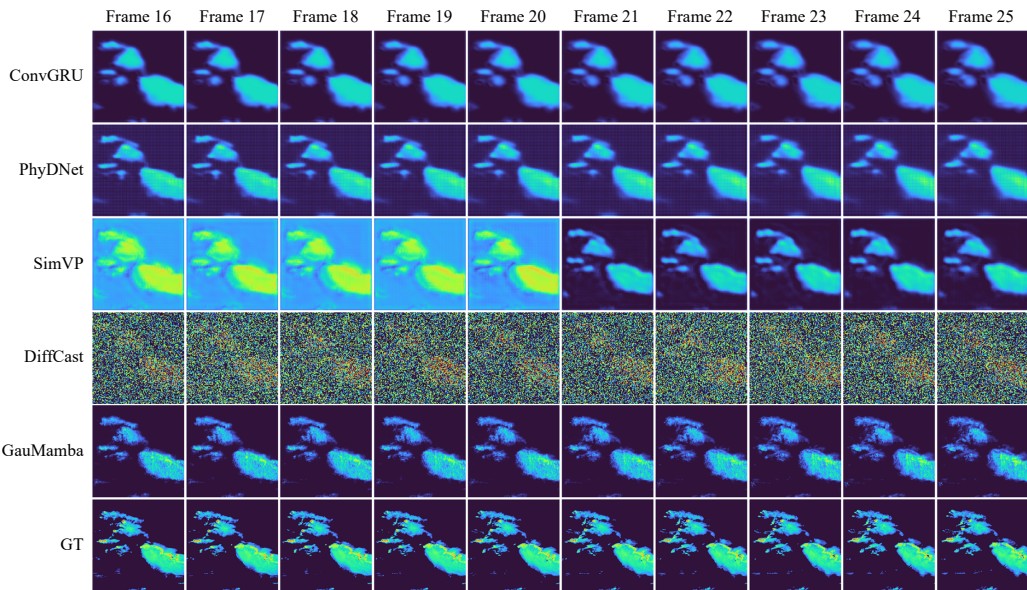

Figure 13: Qualitative results of 3D radar prediction from frame 16 to frame 25 in NEXRAD dataset.

**Discussion of convergence issue in reconstruction.** To accurately reconstruct highly dynamic radar sequences, all parameters of the 3D Gaussians need to be optimized. Following this setting the existing dynamic Gaussian reconstruction methods face significant challenges to convergence. The fixed parameters serve as efficient anchors, allowing each Gaussian to lock onto a corresponding region of the object being reconstructed, thereby consistently tracking changes in the object. When all parameters become adjustable, this crucial prior constraint is lost, leading to incoherent movement of the Gaussians with the reconstructed parts, and resulting in a vague and suboptimal optimization objective. In an extreme case, 3D Gaussians exhibiting random Brownian motion can still reconstruct dynamic scenes and achieve a suboptimal outcome, as long as the deformation function or neural network used for reconstruction has sufficient capacity and enough iterations. Furthermore, each

Gaussian influences only a very small region of the space, which theoretically limits the effective area for gradient descent. A small perturbation introduced by randomly initialized neural networks or polynomial functions can cause the Gaussian to jump out of its original reconstruction region, hindering the gradient flow from the corresponding ground truth area from properly affecting the disturbed Gaussian and trapping it in an unrelated region, leading to significant convergence issues.

