# OpenReview forum: "High-Dynamic Radar Sequence Prediction for Weather Nowcasting Using Spatiotemporal Coherent Gaussian Representation"
_ICLR.cc/2025/Conference — ICLR 2025 Oral_

### Official Review · Reviewer_U22W · 2024-10-28

**Soundness:** 3
**Presentation:** 3
**Contribution:** 2
**Rating:** 8
**Confidence:** 4

**Summary:**

This paper presents a comprehensive framework for 3D radar sequence prediction in weather nowcasting that combines a novel SpatioTemporal Coherent Gaussian Splatting (STC-GS) representation with a memory-augmented predictive network (GauMamba). The approach addresses the key limitations of current 2D prediction methods by enabling efficient and accurate 3D sequence prediction while maintaining computational efficiency.

The main contributions include:

1. A novel 3D radar representation method (STC-GS) that efficiently captures radar data dynamics through bidirectional reconstruction with dual-scale constraints, achieving 16× higher spatial resolution compared to existing 3D representation methods

2. A memory-augmented predictive model (GauMamba) that effectively learns temporal evolution patterns from the STC-GS representations to forecast radar changes, outperforming state-of-the-art methods in prediction accuracy

3. Two new high-dynamic 3D radar sequence datasets:
- MOSAIC: 24,542 single-channel high-resolution radar observations
- NEXRAD: 6,255 six-channel radar observations

**Strengths:**

Good Originality:
- The paper presents a novel solution for 3D radar sequence prediction by combining Gaussian Splatting with a memory-augmented network
- The adaptation of 3D Gaussian Splatting to dynamic radar data representation represents meaningful innovation in both representation and prediction aspects
- The bidirectional reconstruction pipeline with dual-scale constraints is a creative approach to handle the unique challenges in radar sequence prediction

Good Quality:
- The technical development is thorough with comprehensive theoretical foundations and implementation details
- The experimental evaluation is extensive, covering multiple datasets and comparing with various baselines
- The ablation studies effectively validate the contribution of each component

Good Clarity:
- The paper is well-structured with clear motivation and problem formulation
- The methodology is presented in a logical flow with detailed explanations
- The figures are well-designed and effectively illustrate the key concepts
- The writing is generally clear and easy to follow

Good Significance:
- The work addresses an important practical problem in weather nowcasting
- The proposed framework achieves significant improvements over existing methods:
  * 16× higher spatial resolution in representation
  * 19.7% and 50% reduction in MAE on two datasets
- The introduction of two new high-dynamic 3D radar sequence datasets contributes valuable resources to the research community

> While I support accepting this paper based on its technical merits and clear presentation, I am not very familiar in radar sequence prediction specifically. Therefore, I remain open to adjusting my assessment during the discussion phase based on comments from other reviewers more specialized in this domain.

**Weaknesses:**

Major Weaknesses: about **Reconstruction Results**

1. Unusual Performance Gap in Reconstruction:
- The results in Table 1 show dramatically better performance compared to baselines
- While authors explain this is due to convergence issues in existing methods, this raises concerns:
  * The performance gap (10× improvement in MAE) seems unusually large
  * Need to consider whether alternative baselines outside 3DGS family might be more appropriate
  * Traditional scene reconstruction methods might provide more reasonable comparisons

> Therefore, I would like to suggest the authors to include non-3DGS based methods that might achieve better convergence

2. Missing Qualitative Results:
- The paper lacks visualization results for the reconstruction stage
- This omission is particularly concerning given the significant quantitative improvements claimed
- Visual results would help validate the dramatic performance improvements

> Therefore, I would like to suggest the authors to either:
>   * Add qualitative reconstruction visualizations
>   * OR Provide clear justification for why such visualizations are not included


Minor Weaknesses: Grammar and Writing Issues

- Several grammatical errors throughout the paper
    - For example, on Page 7, line 371

> These should be carefully corrected in the final version

**Questions:**

Please see Weaknesses

---

> ### Author Response · Authors · 2024-11-23
> **Response to U22W (1/2)**
>
> Thank you for your detailed and constructive feedback. We greatly appreciate your insights, which have helped us identify areas for improvement in our work.
>
> **[W1]** Performance gap in reconstruction
>
> **A:** We appreciate the reviewer’s observation regarding the large performance gap between our method and the baselines in *Table 1*. To clarify, in our original experiments, we ensured fair comparisons by aligning all methods under a unified reconstruction setting. Specifically, we allowed **all parameters to be freely optimized**, as in our proposed method. However, baseline methods were originally designed with **some parameters fixed** (e.g., RGB or opacity), which serve as an implicit anchor to guide the alignment of Gaussians with the dynamic changes of the reconstruction target. This guidance mechanism is absent in our unified setting, inherently making the reconstruction task more challenging.
>
> Without this guidance, optimizers may struggle to identify which parameters to adjust for better dynamic reconstruction. This leads to convergence issues. To further investigate this phenomenon, we conducted additional experiments where we compared the baselines both **in their original settings** (with fixed parameters) and **in our unified setting**. The updated results are summarized below:
>
> | Model | MAE$^\downarrow$ | PSNR(dB)$^\uparrow$ | SSIM$^\uparrow$ | LPIPS$^\downarrow$ |
> | :------------ | :------------: | :---------------: | :-----------: | :--------------: |
> | 3DGStream$_\text{our}$ | 0.0210 | 14.451 | 0.818 | 0.228 |
> | 4D-GS$_\text{our}$ | 0.0331 | 19.178 | 0.172 | 0.317 |
> | Deform 3DGS$_\text{our}$ | 0.0115 | 26.218 | 0.543 | 0.194 |
> | 3DGStream$_\text{origin}$ | 0.0019 | 38.133 | 0.954 | 0.091 |
> | 4D-GS$_\text{origin}$ | 0.0028 | 35.731 | 0.933 | 0.135 |
> | Deform 3D-GS$_\text{origin}$ | 0.0029 | 35.027 | 0.931 | 0.141 |
> | Ours | **0.0014** | **40.262** | **0.970** | **0.057** |
>
> As demonstrated, when the baselines adhere to their original settings, the performance gap narrows significantly. However, in the unified setting, baseline methods face convergence issues due to the increased complexity of the optimization task. Our proposed strategy efficiently handles these challenges, achieving superior performance.
>
> Regarding the suggestion to include baselines outside the 3DGS family: recent studies have consistently demonstrated that 3D Gaussian-based methods outperform earlier approaches such as NeRF and voxel grid representations, particularly in dynamic scene reconstruction tasks. A key distinction lies in the nature of these methodologies: 3D Gaussian-based techniques focus on **explicit reconstructions**, providing a direct and interpretable representation of the scene, while methods like NeRF and voxel grids adopt **implicit reconstruction** frameworks, relying on latent representations that are less suited for explicit scene modeling and manipulation. Incorporating baselines from outside the 3DGS family would introduce discrepancies in objectives and compatibility, potentially diluting the focus of our study.
>
> In the context of our work, which is designed to leverage the advantages of 3D Gaussian representations for both reconstruction and subsequent prediction. This experimental design aims to emphasize the effectiveness of our reconstruction strategy in challenging scenarios based on 3DGS techniques. Demonstrating improvements over existing 3DGS methods highlights the robustness and adaptability of our approach.
>
> We hope this clarification and additional experimental evidence address the reviewer’s concerns. Thank you again for the constructive feedback.

---

> ### Author Response · Authors · 2024-11-23
> **Response to U22W (2/2)**
>
> **[W2]** Qualitative results for reconstruction stage
>
> **A:** Thank you for your constructive feedback. We appreciate your suggestion regarding the inclusion of visualization results. We have added qualitative visualization results for the reconstruction stage in the main paper. Additionally, we have included comprehensive visual comparisons in the supplementary material to further validate the performance improvements claimed in the paper.
>
> These visualizations clearly demonstrate that our proposed reconstruction method preserves significantly **more details** and achieves **consistent accuracy** across all frames. In contrast, other methods exhibit noticeable deviations in the reconstructed structure by the final frame, such as incorrect patterns or blurred regions. These qualitative results further validate the dramatic performance improvements claimed in the paper.
>
> **[W3]** Grammar and Writing Issues
>
> **A:** Thank you for your careful review. We have thoroughly rechecked the entire manuscript and corrected grammatical errors to improve the overall readability and quality of the paper.

---

> > ### Comment · Reviewer_U22W · 2024-11-24
> >
> > Thank you for providing such a thorough and detailed response. Your explanations have effectively addressed all my previous concerns: the comprehensive analysis of reconstruction performance differences is convincing, the added visualization results effectively validate the quantitative metrics. Besides, the commitment to releasing the datasets will bring significant value to the research community.
> >
> > Based on your response, I am fully convinced of both the technical contributions and the thoroughness of the experimental validation. Therefore, I am pleased to raise my assessment of this paper.

---

> > > ### Author Response · Authors · 2024-11-27
> > >
> > > Thank you for your thoughtful feedback and positive reassessment of our work. We appreciate your recognition of the technical contributions, experimental validation, and the value of our dataset release. Your encouraging remarks are a great motivation, and we are grateful for your support in enhancing the quality of our paper.

---

### Official Review · Reviewer_yyVm · 2024-11-02

**Soundness:** 3
**Presentation:** 3
**Contribution:** 3
**Rating:** 8
**Confidence:** 4

**Summary:**

The paper proposed a framework utilizing SpatioTemporal Coherent Gaussian Splatting (STC-GS) for dynamic radar representation and GauMamba for forecasting dynamic meteorological radar signals. The STC-GS established 4D dynamic Gaussians, and the GauMamba utilizes the Mamba framework to predict 3D radar sequences. The experiments on NEXRAD and MOSAIC present superior performance against previous 4D reconstruction methods.

**Strengths:**

- The writing is comprehensive and easy to follow

- The ablation study is thorough and provides insights into the model design and the selection of hyperparameters.

- By combining Gaussian and Mamba methodologies, the GauMamba model is likely designed to enhance forecasting accuracy, especially in scenarios involving temporal and spatial data complexities which are typical in meteorological datasets.

**Weaknesses:**

- Previous research has introduced combinations such as Gamba that combine Gaussian and Mamba model. However, this paper does not sufficiently review these previous works. It would be beneficial to highlight how the proposed GauMamba method differs from earlier studies. Additionally, including comparisons with Mambda + Gaussian baseline methods in the experimental section is crucial for a sufficient comparisons.

- The paper describes experiments conducted on two weather forecasting datasets. These datasets appear to be quite limited, containing only two scenes—if this is incorrect, please advise. Such a small dataset may not adequately demonstrate how the method compares with established baselines. Conducting experiments on larger, more diverse datasets could provide a more thorough evaluation.

References:
- Shen, Qiuhong, et al. "Gamba: Marry gaussian splatting with mamba for single view 3d reconstruction." arXiv preprint arXiv:2403.18795 (2024).

**Questions:**

- The paper mentions employing a group of Gaussian primitives for radar sequence prediction. Is there a fixed number of Gaussian primitives used across all frames, and if so, how is this number determined and maintained throughout the model's operations?

- It is claimed that the GauMamba model is efficient; however, from the comparisons in Figure 4, it appears that at resolutions lower than 256, baseline methods are more memory efficient. Please discuss why there is this discrepancy in memory efficiency at different resolutions and how the baselines' memory scales with increasing resolution.

- Please also discuss the limitations of the proposed method.

**Details Of Ethics Concerns:**

No ethics concerns.

---

> ### Author Response · Authors · 2024-11-22
> **Response to yyVm (1/4)**
>
> Thank you for your valuable feedback and thoughtful suggestions. We appreciate the opportunity to improve our paper based on your comments. Below, we provide detailed responses to each point raised.
>
> ***
>
> **[W1]** Review and compare with previous Mambda + Gaussian methods
>
> **A:** We sincerely thank the reviewer for the valuable suggestion. To address this, we plan to expand the **Related Work** section by briefly reviewing existing methods that combine Mamba and Gaussian models and highlighting the distinctions of our proposed GauMamba method. Specifically, we will add the following content to Section 2.1:
>
> > Several studies [1-4] have attempted to integrate Mamba or Transformer architectures with 3D Gaussian. However, these methods primarily focus on reconstructing 3D Gaussians from single or multi-view images. In contrast, our method emphasizes utilizing sequences of 3D Gaussians to represent the evolution of 3D radar echo sequences, and employing the GauMamba model to predict future frames. Existing methods lack the capacity to retain past observations. In contrast, our proposed **Memory-Augmented GauMamba** effectively incorporates observations from preceding frames to model the spatiotemporal evolution of 3D Gaussians, significantly improving prediction accuracy.
>
> Additionally, we have included new comparisons in the **Experiment** section with the original Mamba-based model as a baseline. The results are presented in the following table:
>
> **Table 1** *Experiment results in MOSAIC:*
> | Model | ME$^{\rightarrow 0}$ | MAE$^\downarrow$ | SSIM$^\uparrow$ | LPIPS$^\downarrow$ | LPIPS$_{\text{Radar}}^\downarrow$ | CSI-20$_{\text{Pool}}^\uparrow$ | CSI-30$_{\text{Pool}}^\uparrow$ | CSI-40$_{\text{Pool}}^\uparrow$ |
> | :------: | :----------------: | :------------: | :-----------: | :--------------: | :-----------------------------: | :----------------------------: | :----------------------------: | :----------------------------: |
> | ConvGRU | -0.122 | 1.728 | 0.621 | 0.303 | 4.837 | - | - | - |
> | PhyDNet | 0.151 | 0.910 | 0.810 | 0.244 | 1.451 | 0.294 | 0.108 | 0.002 |
> | SimVP | 0.105 | 0.890 | 0.835 | 0.270 | 3.516 | 0.264 | 0.075 | - |
> | DiffCast | 1.092 | 1.878 | 0.355 | 0.433 | 2.216 | 0.305 | 0.126 | 0.006 |
> | Mamba | -0.367 | 0.750 | 0.894 | 0.164 | 0.777 | 0.293 | 0.166 | 0.055 |
> | GauMamba | **-0.103** | **0.714** | **0.897** | **0.157** | **0.741** | **0.342** | **0.213** | **0.062** |
>
> **Table 2** *Experiment results in NEXRAD:*
> | Model | ME$^{\rightarrow 0}$ | MAE$^\downarrow$ | SSIM$^\uparrow$ | LPIPS$^\downarrow$ | LPIPS$_{\text{Radar}}^\downarrow$ | CSI-20$_{\text{Pool}}^\uparrow$ | CSI-30$_{\text{Pool}}^\uparrow$ | CSI-40$_{\text{Pool}}^\uparrow$ |
> | :------: | :----------------: | :------------: | :-----------: | :--------------: | :-----------------------------: | :----------------------------: | :----------------------------: | :----------------------------: |
> | ConvGRU | 0.0008 | 0.006 | 0.819 | 0.205 | 1.621 | 0.306 | - | - |
> | PhyDNet | 0.0139 | 0.017 | 0.373 | 0.320 | 2.058 | 0.311 | 0.089 | 0.002 |
> | SimVP | 0.0230 | 0.066 | 0.379 | 0.481 | 2.925 | 0.085 | 0.088 | 0.018 |
> | DiffCast | 0.1525 | 0.157 | 0.004 | 0.932 | 4.057 | 0.049 | 0.021 | 0.021 |
> | Mamba | -0.0016 | 0.004 | 0.899 | 0.129 | 0.699 | 0.309 | 0.165 | 0.074 |
> | GauMamba | **0.0006** | **0.003** | **0.900** | **0.126** | **0.665** | **0.326** | **0.179** | **0.078** |
>
> We would like to emphasize that the **original Mamba model lacks memory capabilities**, meaning it cannot retain information from previous frames and **can only predict the next frame based on the current state**. The comparison with GauMamba clearly demonstrates that the original Mamba performs suboptimally in spatiotemporal sequence tasks.
>
> More importantly, the **approaches and experiements developed within our proposed framework**, which reformulates the 3D radar sequence prediction task by first re-representing the sequence with 3D Gaussians and then predicting future frames, **significantly outperform traditional methods**. This underscores the effectiveness of our novel framework. Additionally, our Memory-Augmented Mamba Predictive Model further enhances the predictive power of Mamba-based models.
>
> ***References:***
>
> [1] Shen, Qiuhong, et al. "Gamba: Marry gaussian splatting with mamba for single view 3d reconstruction." arXiv preprint arXiv:2403.18795 (2024).
>
> [2] Yi, Xuanyu, et al. "MVGamba: Unify 3D Content Generation as State Space Sequence Modeling." arXiv preprint arXiv:2406.06367 (2024).
>
> [3] Ziwen, Chen, et al. "Long-lrm: Long-sequence large reconstruction model for wide-coverage gaussian splats." arXiv preprint arXiv:2410.12781 (2024).
>
> [4] Zhang, Kai, et al. "Gs-lrm: Large reconstruction model for 3d gaussian splatting." European Conference on Computer Vision. Springer, Cham, 2025.

---

> ### Author Response · Authors · 2024-11-22
> **Response to yyVm (2/4)**
>
> **[W2]** Dataset size and diversity
>
> **A:** Thank you for raising this important concern. Our experiments were conducted on two 3D radar sequence datasets: **MOSAIC** and **NEXRAD**.
>
> - **Dataset diversity:**
>
>     **MOSAIC** consists of a full year of radar observations from the region spanning the intersection of Northeast Asia and Southeast Asia, near the western Pacific. This region exhibits significant climatic diversity due to its unique geographical and meteorological conditions. It experiences **a wide range of weather phenomena** such as convective rain, stratiform precipitation, monsoons, and typhoons throughout the year, making it an ideal dataset to capture various weather dynamics.
>
>     **NEXRAD** comprises radar data collected from multiple observation stations across the United States. This dataset reflects not only **climatic diversity** but also **geographical variability**, as it covers regions including such as mountains, plains, and urban areas.
>
>     Such variability enhances the robustness of our method's evaluation.
>
> - **Dataset size:**
>
>     We acknowledge the broader challenge of limited availability of diverse 3D radar datasets. Several factors contribute to this limitation:
>
>     - **Technical constraints:** Many radar systems cannot reliably capture accurate 3D observations.
>
>     - **Research focus:** Current meteorological studies often prioritize 2D low-altitude radar imagery, with the collection and utilization of 3D radar data still in exploratory stages.
>
>     - **Data access restrictions:** Stringent security and privacy policies in many countries prevent public sharing of 3D radar data.
>
>     Despite these challenges, the NEXRAD dataset, derived from the U.S. NEXRAD WSR-88D radar network, is widely regarded in the remote sensing community for its reliability and representativeness. Moreover, the combination of NEXRAD with MOSAIC dataset introduces complementary perspectives, as these datasets stem from distinct meteorological systems and utilize different radar equipment, enhancing the diversity of our experimental setup.
>
>     To further address concerns about dataset size, **we are extending our experiments by incorporating additional radar data from 2020 and 2021**. This expansion aims to evaluate whether increasing the dataset size further enhances model accuracy. As the experiment is being conducted, we plan to update the results in the next few days.
>
>     Additionally, we plan to publicly release this three-year dataset to support future research in the community.

---

> ### Author Response · Authors · 2024-11-22
> **Response to yyVm (3/4)**
>
> **[Q1]** A fixed number of Gaussian primitives
>
> **A:** Thanks for your detailed question. **Yes**, the number of Gaussian primitives is fixed across all frames in our model. This setting is motivated by the necessity to ensure spatiotemporal consistency during sequence reconstruction and reparameterization. Dynamically adding or removing Gaussian primitives would disrupt this consistency, and make it challenging for the model to capture the temporal evolution of each individual 3D Gaussian.
>
> Our approach aligns with findings in Taming 3DGS [1] which highlights that the densification operations in the original 3D Gaussian Splatting (3DGS) framework may introduce challenges for subsequent training. Similarly, many studies [2-7] incorporating 3D Gaussians into deep learning frameworks also adopt a fixed number of primitives for training simplicity and stability.
>
> In our implementation, we set the number of Gaussian primitives to $3 \times 2^{14}$, balancing computational efficiency and reconstruction precision. During initialization, these primitives are randomly distributed within regions containing valid radar echoes. To ensure their effectiveness, we propose a **Bidirectional Reconstruction Scheme** coupled with **local detail and global trend constraints**, which enables each primitive to meaningfully contribute to the reconstruction process and remain aligned with the motion of the corresponding cloud structures.
>
> We appreciate your feedback and hope this explanation addresses your concerns.
>
> ***Refereces:***
>
> [1] Mallick, Saswat Subhajyoti, et al. "Taming 3dgs: High-quality radiance fields with limited resources." arXiv preprint arXiv:2406.15643 (2024).
>
> [2] Shen, Qiuhong, et al. "Gamba: Marry gaussian splatting with mamba for single view 3d reconstruction." arXiv preprint arXiv:2403.18795 (2024).
>
> [3] Yi, Xuanyu, et al. "MVGamba: Unify 3D Content Generation as State Space Sequence Modeling." arXiv preprint arXiv:2406.06367 (2024).
>
> [4] Ziwen, Chen, et al. "Long-lrm: Long-sequence large reconstruction model for wide-coverage gaussian splats." arXiv preprint arXiv:2410.12781 (2024).
>
> [5] Zhang, Kai, et al. "Gs-lrm: Large reconstruction model for 3d gaussian splatting." European Conference on Computer Vision. Springer, Cham, 2025.
>
> [6] Tang, Jiaxiang, et al. "Lgm: Large multi-view gaussian model for high-resolution 3d content creation." European Conference on Computer Vision. Springer, Cham, 2025.
>
> [7] Lu, Guanxing, et al. "Manigaussian: Dynamic gaussian splatting for multi-task robotic manipulation." European Conference on Computer Vision. Springer, Cham, 2025.
>
> **[Q2]** GauMamba's memory efficiency
>
> **A:** Thank you for raising this insightful question. The observed discrepancy in memory efficiency at different resolutions arises from the fundamental differences between the underlying mechanisms of the baseline methods and our GauMamba model.
>
> The baseline methods rely on **convolutional architectures**, where memory usage is directly proportional to the size of the feature maps. This results in **quadratic growth** in memory consumption as the spatial resolution increases (considering horizontal dimensions). In contrast, our approach employs a fixed number of Gaussian primitives, making memory usage **independent of the resolution**. Instead, memory consumption in our model **scales linearly** with the number of Gaussian primitives.
>
> At lower resolutions, most radar echo details are lost, which can significantly reduce the practical utility of the predictions. For example, many intense meteorological events evolve from small-scale structures, and low-resolution models may fail to capture these early-stage developments. Additionally, low-resolution predictions can overly average regions of high reflectivity, obscuring critical localized features. High-resolution predictions, on the other hand, provide more detailed guidance for disaster prevention and mitigation, enabling timely and precise decision-making.
>
> Furthermore, the number of Gaussian primitives in our method is tailored to the specific resolution of the radar data. Reconstructing low-resolution radar sequences requires fewer Gaussian primitives to represent the underlying features. Therefore, at low resolutions, the reduced number of Gaussian primitives also leads to higher memory efficiency.
>
> In *Figure 4*, we aimed to highlight that under the current experimental settings, our approach demonstrates superior memory efficiency, particularly at higher resolutions where practical applications are most relevant.

---

> ### Author Response · Authors · 2024-11-22
> **Response to yyVm (4/4)**
>
> **[Q3]** Limitations of our method
>
> **A:** Thank you for emphasizing the importance of discussing the limitations of our method.
>
> - **From the application perspective**, while our approach demonstrates strong performance in reconstructing and predicting highly dynamic radar sequences, its current scope is somewhat specialized, focusing primarily on radar-based weather nowcasting and sequence prediction. We acknowledge that broader applications, such as general 3D scene reconstruction and dynamic modeling for robotics or AR/VR environments, have not been explored in this work. Expanding our method into a foundational framework for 3D world modeling is a direction we are actively pursuing.
>
> - **From the technical perspective**, the reconstruction strategy employed in our method, while effective, is somewhat tailored to radar sequences, which may limit its adaptability to other dynamic 3D scenarios, such as autonomous driving, robotic movements or manipulation. To address this, we are working toward developing a more generalized reconstruction framework and simplifying the pipeline to enhance its versatility and computational efficiency across diverse domains.
>
> We'll add a short discussion of limitations to the main paper.

---

> ### Comment · Reviewer_yyVm · 2024-11-24
> **Response to Rebuttal**
>
> I really appreciate the comprehensive explanation and additional comparison with Mamba. My concerns have been sufficiently addressed, so I raised my score to 8 (accept, good paper). I encourage the authors to consider making their dataset publicly available in the future, as it would significantly benefit the fields of weather forecasting and 4D reconstruction.

---

> > ### Author Response · Authors · 2024-11-27
> >
> > Thank you very much for your positive feedback and for raising the score to 8. We truly appreciate your recognition of our work and your constructive suggestions.
> >
> > * We are pleased to share that we have already released a 3-year NEXRAD dataset, which can be accessed via https://huggingface.co/datasets/Ziyeeee/3D-NEXRAD. We warmly welcome researchers to utilize this dataset and our framework for further exploration in weather forecasting and 4D reconstruction.
> >
> > * Additionally, as suggested in **[W2]**, we conducted extended experiments using this dataset. The results are in the table below:
> > | Model | MAE$^\downarrow$ | SSIM$^\uparrow$ | LPIPS$^\downarrow$ | LPIPS$_\text{Radar}^\downarrow$ | CSI-20$_{\text{Pool4}}^\uparrow$ | CSI-30$_{\text{Pool4}}^\uparrow$ | CSI-40$_{\text{Pool4}}^\uparrow$ |
> > | :------------: | :------------: | :------------: | :------------: | :------------: | :------------: | :------------: | :------------: |
> > | ConvGRU | 0.006 | 0.836 | 0.194 | 1.632 | 0.326 | - | - |
> > | PhyDNet | 0.017 | 0.366 | 0.323 | 2.114 | 0.348 | 0.097 | 0.002 |
> > | SimVP | 0.008 | 0.817 | 0.176 | 1.483 | 0.227 | 0.002 | 0.000 |
> > | DiffCast | 0.152 | 0.005 | 0.925 | 4.005 | 0.051 | 0.023 | 0.044 |
> > | Mamba | 0.004 | 0.902 | 0.125 | 0.625 | 0.304 | 0.158 | 0.075 |
> > | GauMamba | **0.003** | **0.907** | **0.122** | **0.600** | **0.361** | **0.205** | **0.089** |
> >
> >     Most models demonstrate slight improvements in metrics when trained on the extended dataset. This performance enhancement can be attributed to the increased diversity of data and the additional iteration steps allowed by the larger dataset size, as the same number of epochs was maintained. Notably, our model consistently outperforms others, highlighting its robustness and effectiveness even under these extended experimental conditions.
> >
> >     We have incorporated these results and the corresponding discussion into the revised version of the manuscript.
> >
> > Thank you again for your valuable feedback, which has significantly contributed to improving the quality of our paper.

---

### Official Review · Reviewer_pwg9 · 2024-11-04

**Soundness:** 3
**Presentation:** 3
**Contribution:** 3
**Rating:** 8
**Confidence:** 2

**Summary:**

This paper presents a novel 3D weather nowcasting approach using high-dynamic radar sequences. The method introduces a SpatioTemporal Coherent Gaussian Splatting technique to efficiently represent the dynamic radar data, which is then processed by the GauMamba network to generate weather forecasts. Additionally, the paper proposes MOSAIC, a new high-resolution 3D radar sequence dataset, containing more than 24K radar observations. Experimental results on both NEXRAD and MOSAIC show that the proposed approach outperforms baseline methods with significant margins.

**Strengths:**

- The proposed representation and processing pipeline are well-motivated and supported by the experiments, resulting in performance improvements over baseline methods.
- The memory usage of the method remains constant w.r.t. horizontal resolution, in contrast to other baselines with linear memory growth.
- The MOSAIC dataset offers a large dataset of radar echoes that capture meteorological events across multiple years.

**Weaknesses:**

- A short discussion of the proposed dataset, MOSAIC, in the main paper could be beneficial to the readers.
- Could the authors clarify the plan for the dataset? Will it be made publicly available, and if so, how to ensure that the dataset can be accessed by the public continuously?
- For the LPIPS evaluation, how is the evaluator model trained? If the model is pretrained with a general-purpose dataset, would it work well with radar data?
- Minor visual artifacts can be seen in Figure 2 (center bottom).

**Questions:**

Please see Weaknesses.

---

> ### Author Response · Authors · 2024-11-22
> **Response to pwg9 (1/3)**
>
> Thank you for your insightful and constructive feedback. We greatly appreciate your comments and have carefully addressed each point in our detailed responses below.
>
> ***
>
> **[W1]** _A short discussion of the proposed dataset_
>
> **A:** Thank you for your valuable suggestion. Including a brief discussion about the two dataset in the main paper will benefit readers. We will add the following content to Section 4.1:
>
> > The datasets used in this study include NEXRAD and MOSAIC. NEXRAD comprises 6255 radar observations of severe storms in the U.S., with 3D reflectivity data sampled at 5-minute intervals and a resolution of $ 44 \times 512 \times 512 $. Seven radar features, such as reflectivity, azimuthal shear, differential reflectivity, and so on, are included. MOSAIC records 24,542 radar observations of storms in Guangdong, China, with 6-minute intervals and a resolution of $ 38 \times 384 \times 512 $, focusing solely on intensity data of radar echoes. Both datasets are preprocessed to ensure consistent vertical spacing and are divided into training, validation, and test sets. The prediction task involves forecasting up to 20 future frames based on 5 observed frames. For further information, please refer to the supplementary material C.1.
>
> ***
>
> **[W2]** _Clarify the public plan for the dataset_
>
> **A:** Thank you for your question regarding the dataset. We confirm that we plan to make the datasets used in this work publicly available. Specifically, this paper utilizes two datasets: **NEXRAD** and **MOSAIC** .
>
> -  **NEXRAD:** This dataset is derived from the U.S. NEXRAD WSR-88D radar network. We have already uploaded the processed dataset to Hugging Face for public access, and it can be accessed via the following link: https://huggingface.co/datasets/Ziyeeee/3D-NEXRAD. Additionally, we are currently extending this dataset to include data from 2020 and 2021. These years are undergoing final organization and processing and are expected to be released in this month (November 2024).
>
> - **MOSAIC:** This dataset originates from the National Meteorological Centre. Its release requires additional approvals and security reviews. We are actively coordinating with the relevant authorities to expedite the process and will make the dataset available as soon as possible.

---

> ### Author Response · Authors · 2024-11-22
> **Response to pwg9 (2/3)**
>
> **[W3]** _The evaluator model of LPIPS and a pretrained model with radar data_
>
> **A:** Thank you for your insightful question. We employed a pretrained **AlexNet** model for the LPIPS evaluation, following the settings in the original 3D Gaussian Splatting (3DGS) paper [1] and the suggestion in LPIPS framework [2]. We will clarify this detail in the revised manuscript.
>
> Regarding your concern about whether a model pretrained on a general-purpose dataset is suitable for radar data, we agree that this is an interesting and important question. To address this, we conducted additional experiments. One notable challenge in radar data lies in the scarcity of labeled datasets for supervised training. However, as highlighted in [2], self-supervised models like BiGAN and supervised AlexNet calibrated with human perceptual judgments achieve comparable performance in measuring perceptual distance (68.4 vs. 69.8 2AFC scores). **This indicates that the evaluator model does not necessarily need to be trained on a classification task to perform effectively; self-supervised models can achieve results on par with supervised ones.**
>
> Based on this observation, **we pretrained a BiGAN model on radar data in a self-supervised manner** and used its encoder as the evaluator for LPIPS. This approach resulted in the radar-specific perceptual metric, $\text{LPIPS}_{\text{Radar}}$, as shown in the updated tables below:
>
> **Table 1** *Experiment results in MOSAIC:*
> | Model | ME$^{\rightarrow 0}$ | MAE$^\downarrow$ | SSIM$^\uparrow$ | LPIPS$^\downarrow$ | LPIPS$_{\text{Radar}}^\downarrow$ | CSI-20$_{\text{Pool}}^\uparrow$ | CSI-30$_{\text{Pool}}^\uparrow$ | CSI-40$_{\text{Pool}}^\uparrow$ |
> | :------: | :----------------: | :------------: | :-----------: | :--------------: | :-----------------------------: | :----------------------------: | :----------------------------: | :----------------------------: |
> | ConvGRU | -0.122 | 1.728 | 0.621 | 0.303 | 4.837 | - | - | - |
> | PhyDNet | 0.151 | 0.910 | 0.810 | 0.244 | 1.451 | 0.294 | 0.108 | 0.002 |
> | SimVP | 0.105 | 0.890 | 0.835 | 0.270 | 3.516 | 0.264 | 0.075 | - |
> | DiffCast | 1.092 | 1.878 | 0.355 | 0.433 | 2.216 | 0.305 | 0.126 | 0.006 |
> | Mamba | -0.367 | 0.750 | 0.894 | 0.164 | 0.777 | 0.293 | 0.166 | 0.055 |
> | GauMamba | **-0.103** | **0.714** | **0.897** | **0.157** | **0.741** | **0.342** | **0.213** | **0.062** |
>
> **Table 2** *Experiment results in NEXRAD:*
> | Model | ME$^{\rightarrow 0}$ | MAE$^\downarrow$ | SSIM$^\uparrow$ | LPIPS$^\downarrow$ | LPIPS$_{\text{Radar}}^\downarrow$ | CSI-20$_{\text{Pool}}^\uparrow$ | CSI-30$_{\text{Pool}}^\uparrow$ | CSI-40$_{\text{Pool}}^\uparrow$ |
> | :------: | :----------------: | :------------: | :-----------: | :--------------: | :-----------------------------: | :----------------------------: | :----------------------------: | :----------------------------: |
> | ConvGRU | 0.0008 | 0.006 | 0.819 | 0.205 | 1.621 | 0.306 | - | - |
> | PhyDNet | 0.0139 | 0.017 | 0.373 | 0.320 | 2.058 | 0.311 | 0.089 | 0.002 |
> | SimVP | 0.0230 | 0.066 | 0.379 | 0.481 | 2.925 | 0.085 | 0.088 | 0.018 |
> | DiffCast | 0.1525 | 0.157 | 0.004 | 0.932 | 4.057 | 0.049 | 0.021 | 0.021 |
> | Mamba | -0.0016 | 0.004 | 0.899 | 0.129 | 0.699 | 0.309 | 0.165 | 0.074 |
> | GauMamba | **0.0006** | **0.003** | **0.900** | **0.126** | **0.665** | **0.326** | **0.179** | **0.078** |
>
> The results demonstrate that LPIPS_Radar, derived from the BiGAN evaluator, is **well-aligned** with the original LPIPS results. More importantly, **it highlights perceptual differences that were previously undetected by the original LPIPS**, such as a difference between 0.126 and 0.129 in LPIPS, but a more significant difference between 0.665 and 0.699 in LPIPS_Radar.
>
> Notably, in *Table 1*, the scores for ConvGRU and DiffCast show discrepancies between LPIPS and LPIPS_Radar. Referring to *Figure 6 left*, it can be clearly observed that ConvGRU fails to predict the next few frames accurately, instead providing a smoothed average result. In contrast, DiffCast produces results that are closer to the ground truth but with some noise. The higher LPIPS score for DiffCast indicates that LPIPS lacks robustness to noise in radar data, while LPIPS_Radar more accurately reflects the perceptual differences between the two methods. **This tailored evaluator not only better suits the unique characteristics of radar data but also enhances the robustness of our evaluation.** This new method provides a more accurate perception of radar data and its model predictions.
>
> Additionally, in response to Reviewer yyVm’s suggestion, we have included a new comparison with the original Mamba model in the updated table, highlighting the improvements brought by our GauMamba model.
>
> We appreciate your valuable suggestions and will incorporate these results and discussions into the revised manuscript.

---

> ### Author Response · Authors · 2024-11-22
> **Response to pwg9 (3/3)**
>
> (Continuation from above)
>
> ***References:***
>
> [1] Kerbl, Bernhard, et al. "3D Gaussian Splatting for Real-Time Radiance Field Rendering." ACM Trans. Graph. 42.4 (2023): 139-1.
>
> [2] Zhang, Richard, et al. "The unreasonable effectiveness of deep features as a perceptual metric." Proceedings of the IEEE conference on computer vision and pattern recognition. 2018.
>
> **[W4]** *Minor visual artifacts in Figure 2*
>
> **A:** Thanks for your suggestion to improve clarity. The semi-transparent design was originally intended to illustrate that the data for the $t+1$ frame flows through the network in the same manner as the $t$ frame. However, we realize that this design might cause confusion.
>
> To address this, we have removed the semi-transparent elements. Instead, we have provided a clear explanation in the figure caption.

---

### Author Response · Authors · 2024-11-24
**Overall Response**

We sincerely thank all reviewers for their constructive and valuable feedback on our paper.

***

## **In this post:**

1. We summarize the strengths of our paper from the reviewers.

2. We summarize the changes to the updated PDF document.

**In the individual replies,** we address other comments.

***

## **Strengths of Our Paper:**

- **Sound Motivation**

    - **pwg9:** "The proposed representation and processing pipeline are well-motivated."

    - **U22W:** "The adaptation of 3D Gaussian Splatting to dynamic radar data representation represents meaningful innovation in both representation and prediction aspects."

- **Robust Contributions and Insightful Experiments**

    - **pwg9:**
        - "The method demonstrates superior performance over baseline approaches, supported by experiments."
        - "The memory usage of the method remains constant w.r.t. horizontal resolution, in contrast to other baselines with linear memory growth."

    - **yyVM:**
        - "The ablation study is thorough and provides insights into the model design and the selection of hyperparameters."
        - "By combining Gaussian and Mamba methodologies, the GauMamba model is designed to enhance forecasting accuracy, especially in scenarios involving temporal and spatial data complexities."

    - **U22W:**
        - "The bidirectional reconstruction pipeline with dual-scale constraints is a creative approach to handle the unique challenges in radar sequence prediction."
        - "The technical development is thorough with comprehensive theoretical foundations and implementation details."
        - "The experimental evaluation is extensive, covering multiple datasets and comparing with various baselines."
        - "The ablation studies effectively validate the contribution of each component"

- **Significance and Impact**

    - **pwg9:** "The paper proposes MOSAIC, a new high-resolution 3D radar sequence dataset, containing more than 24K radar observations."

    - **U22W:**
        - "The work addresses an important practical problem in weather nowcasting."
        - "The proposed framework achieves significant improvements over existing methods."
        - "The introduction of two new high-dynamic 3D radar sequence datasets contributes valuable resources to the research community."

- **Clarity and Presentation**

    - **yyVM:** "The writing is comprehensive and easy to follow."

    - **U22W:**
        - "The paper is well-structured with clear motivation and problem formulation."
        - "The writing is generally clear and easy to follow."
        - "The methodology is presented in a logical flow with detailed explanations, figures are well-designed and effectively illustrate the key concepts"

***

## **Changes to PDF:**

We have proofread the paper and added extra experimental results in the revised version (**highlighted in blue**).

**Main text**

- **yyVm:** (Section 2.1) We have reviewed existing methods that combine Mamba and Gaussian and highlighted the distinctions of our GauMamba.

- **pwg9:** (Figure 2) We have removed the semi-transparent elements in Fig. 2 and provided a clear explanation in the caption.

- **pwg9:** (Section 4.1) We have added a brief discussion about the two datasets.

- **U22W:** (Table 1) We have updated the table with experimental results obtained under their original settings.

- **U22W:** (Section 4.2) We have added visualization of the results and related discussions.

- **yyVm:** (Section 4.2) We have reclarified the memory efficiency of GauMamba.

- **yyVm:** (Table 2 and 3) We have updated the results of Mamba obtained within our proposed framework.

- **pwg9:** (Table 1, 2 and 3) We have updated the results evaluated by $\text{LPIPS}_\text{Radar}$.

- **yyVm:** (Section 4.2) We have added the discussions about the comparison results of Mamba and Gaumamba.

- **yyVm:** (Section 5) We have added the limitations of our proposed model.

**Appendix**

Additional experiments, analyses, and discussions have been incorporated in response to the reviewers' suggestions:

- **pwg9:** (Section C.3) We have provided more details and discussions about $\text{LPIPS}_\text{Radar}$.

- **yyVm:** (Section D) We have provided results of extended experiments.

- **U22W:** (Section E.1) We have added the full visualization results of the reconstruction stage.

---

### Meta-Review · Area_Chair_XUax · 2024-12-20

**Metareview:**

This paper proposes a novel framework for 3D weather nowcasting, combining SpatioTemporal Coherent Gaussian Splatting (STC-GS) for dynamic radar representation and GauMamba, a memory-augmented predictive network, for forecasting. The approach efficiently captures and predicts high-dynamic radar sequences. Experimental results on datasets demonstrate that the proposed method significantly outperforms baseline and 4D reconstruction techniques in accuracy and efficiency. All reviewers' acknowledge that the proposed method is novel, makes significant contribution, and achieves the SOTA results.

**Additional Comments On Reviewer Discussion:**

The authors successfully addressed the reviewers' comments during the rebuttal phase by improving the clarity of the writing and methodology description, providing a more comprehensive review of similar methods from the literature, and conducting additional experiments. The reviewers acknowledged that their concerns were effectively adressed.

---

### Decision · Program_Chairs · 2025-01-22

Accept (Oral)